# CRYOSPHERE: SINGLE-PARTICLE HETEROGENEOUS RECONSTRUCTION FROM CRYO EM

**Gabriel Ducrocq**
Division of Statistics and Machine Learning
Linköping University, Linköping, Sweden
gabriel.ducrocq@liu.se

**Lukas Grunewald**
Department of Chemistry
Uppsala University, Uppsala, Sweden
lukas.grunewald@kemi.uu.se

**Sebastian Westenhoff**
Department of Chemistry
Uppsala University, Uppsala, Sweden
sebastian.westenhoff@kemi.uu.se

**Fredrik Lindsten**
Division of Statistics and Machine Learning
Linköping University, Linköping, Sweden
fredrik.lindsten@liu.se

## ABSTRACT

The three-dimensional structure of proteins plays a crucial role in determining their function. Protein structure prediction methods, like AlphaFold, offer rapid access to a protein's structure. However, large protein complexes cannot be reliably predicted, and proteins are dynamic, making it important to resolve their full conformational distribution. Single-particle cryo-electron microscopy (cryo-EM) is a powerful tool for determining the structures of large protein complexes. Importantly, the numerous images of a given protein contain underutilized information about conformational heterogeneity. These images are very noisy projections of the protein, and traditional methods for cryo-EM reconstruction are limited to recovering only one or a few consensus conformations. In this paper, we introduce cryoSPHERE, which is a deep learning method that uses a nominal protein structure (e.g., from AlphaFold) as input, learns how to divide it into segments, and moves these segments as approximately rigid bodies to fit the different conformations present in the cryo-EM dataset. This approach provides enough constraints to enable meaningful reconstructions of single protein structural ensembles. We demonstrate this with two synthetic datasets featuring varying levels of noise, as well as two real dataset. We show that cryoSPHERE is very resilient to the high levels of noise typically encountered in experiments, where we see consistent improvements over the current state-of-the-art for heterogeneous reconstruction.

## 1 INTRODUCTION

Single-particle cryo-electron microscopy (cryo-EM) is a powerful technique for determining the three-dimensional structure of biological macromolecules, including proteins. In a cryo-EM experiment, millions of copies of the same protein are first frozen in a thin layer of vitreous ice and then imaged using an electron microscope. This yields a micrograph: a noisy image containing 2D projections of individual proteins. The protein projections are then located on this micrograph and cut out so that an experiment typically yields $10^4$ to $10^7$ images of size $N_{\text{pix}} \times N_{\text{pix}}$ of individual proteins, referred to as *particles*. Our goal is to reconstruct the possible structures of the proteins given these images. Frequently, proteins are conformationally heterogeneous and each copy represents a different structure. Conventionally, this information has been discarded, and all of the sampled structures were assumed to be in only one or a few conformations (*homogeneous* reconstruction). Here, we would like to recover all of the structures in a *heterogeneous* reconstruction.

Structure reconstruction from cryo-EM presents a number of challenges. First, each image shows a particle in a different, unknown orientation. Second, because of the way the electrons interact with the protein, the spectrum of the images is flipped and reduced. Mathematically, this corresponds to a convolution of each individual image with the Point Spread Function (PSF). Third, the images typically have a very low signal-to-noise ratio (SNR). For these reasons, it is very challenging to perform

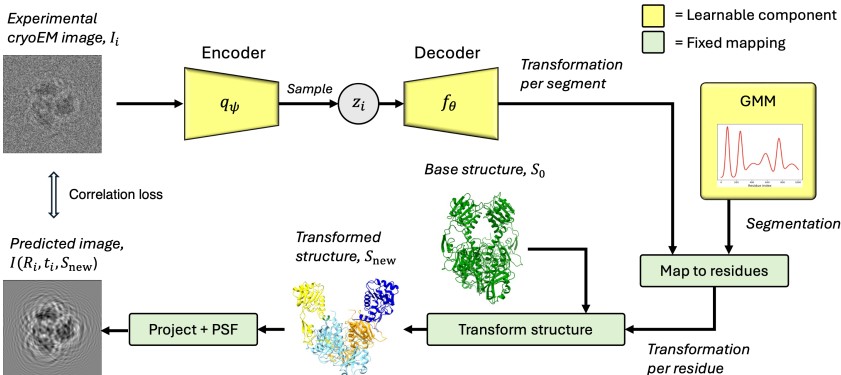

Figure 1: Flow chart of our network. The learnable parts of the model are the encoder, the decoder and the Gaussian mixture. Note that even though the transformations predicted by the decoder are on a per image basis, that is not the case of the Gaussian mixture, which is shared across all particles.

*de novo* cryo-EM reconstruction. Standard methods, produce electron densities averaged over many, if not all conformations (Scheres, 2012; Punjani et al., 2017), performing discrete heterogeneous reconstruction. More recent methods attempt to extract continuous conformational heterogeneity, e.g., by imposing constraints on the problem through an underlying structure deformed to fit the different conformations present in the dataset, see e.g. Rosenbaum et al. (2021); Zhong et al. (2021b); Li et al. (2023). AlphaFold (Jumper et al., 2021) and RosettaFold (Baek et al., 2021) can provide such a structure based on the primary sequence of the protein only. In spite of this strong prior, it is still difficult to recover meaningful conformations. The amount of noise and the fact that we observe only 2D projections creates local minima that are difficult to escape (Zhong et al., 2021b; Rosenbaum et al., 2021), leading to unrealistic conformations.

To remedy this, we root our method in the observation that different conformations can often be explained by large scale movements of domains of the protein (Mardt et al., 2022). Specifically, we develop a variational auto-encoder (VAE) (Kingma & Welling, 2014) that, from a nominal structure and a set of cryo-EM images:

- Learns how to divide the amino-acid chain into segments, given a user defined maximum number of segments; see Figure 2. The nominal structure can for instance be obtained by AlphaFold (Jumper et al., 2021).
- For each image, learns approximately rigid transformations of the identified segments of the nominal structure, which effectively allows us to recover different conformations on an image-by-image (single particle) basis.

These two steps happen concurrently, and the model is end-to-end differentiable. The model is illustrated in Figure 1. The implementation of the model is available on github [1].

Note that what we call a segment is conceptually different from a domain in the structural biology sense. The domains of a protein play a pivotal role in diverse functions, engaging in interactions with other proteins, DNA/RNA, or ligand, while also serving as catalytic sites that contribute significantly to the overall functionality of the protein, see e.g. Schulz & Schirmer (1979); Nelson et al. (2017). By comparison, the segments we learn do not necessarily have a biological function. However, while not strictly necessary for the function of the method, experiments in Section 5 show that our VAE often recovered the actual domains corresponding to different conformations.

## 2 NOTATIONS AND PROBLEM FORMULATION

In what follows, we consider only the $C_\alpha$ atoms of the protein. A protein made of a number $R_{\text{res}} \in \mathbb{N}^\star$ of residues $r_i$ is denoted $S = \{r_i\}_{i=1}^{R_{\text{res}}}$, where the coordinates of residue $i$ are the coordinates of its $C_\alpha$ atom. The electron density map of a structure $S$, also called a volume, is a function $V_S : \mathbb{R}^3 \to \mathbb{R}$, where $V_S(x)$ is proportional to the probability density function of an electron of $S$ being present in an infinitesimal region around $x \in \mathbb{R}^3$. That is, the expected number of electrons in $B \subseteq \mathbb{R}^3$ is proportional to $\int_B V_S(x)\mathrm{d}x$.

---

[1]https://github.com/Gabriel-Ducrocq/cryoSPHERE

Assume we have a set of 2D images $\{I_i\}_{i=1}^N$ of size $N_{\text{pix}} \times N_{\text{pix}}$, representing 2D projections of different copies of the same protein in different conformations. Traditionally, the goal of cryo-EM heterogeneous reconstruction has been to recover, for each image $i$, the electron density map $V_i$ corresponding to the underlying conformation present in image $i$; see Section 4 for a review of these methods. However, following recent works, e.g., Rosenbaum et al. (2021); Zhong et al. (2021b), we aim at recovering, for each image $i$, the underlying structure $S_i$ explaining the image. That is, we try to recover the precise position in $\mathbb{R}^3$ of each residue.

## 3 METHOD – CRYOSPHERE

In this section, we present our method for single-particle heterogeneous reconstruction, denoted cryoSPHERE. The method focuses on structure instead of volume reconstruction. It differs from the previous (Rosenbaum et al., 2021) and concurrent (Li et al., 2023) works along this line in the way the movements of the residues are constrained: instead of deforming the base structure on a residue level and then imposing a loss on the reconstructed structure, our method learns to decompose the amino-acid chain of the protein into segments and, for each image $I_i$, to rigidly move the learnt segments of a base structure $S_0$ to match the conformation present in that image. This is motivated by the fact that different conformations of large proteins often can be explained by large scale movements of its domains (Mardt et al., 2022).

The base structure $S_0$ can be obtained using methods like AlphaFold (Jumper et al., 2021) and RosettaFold (Baek et al., 2021), based on the amino-acid sequence of the protein. In Section 5, we further fit the AlphaFold predicted structure into a volume recovered by a custom backprojection algorithm provided by Zhong et al. (2020).

We use a type of VAE architecture, see Figure 1. We map each image to a latent variable by a stochastic encoder, which is then decoded to a rigid body transformation per segment. Based on these transformations and the segment decomposition, the underlying structure $S_0$ is deformed, posed and turned into a volume that is used to create a projected image. This image is then compared to the input image. After that, the backward pass updates the parameters of the encoder, decoder and Gaussian mixture. We now describe the details of our model.

### 3.1 IMAGE FORMATION MODEL

To compute the 2D projection of the protein structure $S$, we first estimate its 3D electron density map $V$:

$$V_S(r) := \sum_{a \in S} A_a \exp\left(-\frac{||r - a||^2}{2\sigma^2}\right) \tag{1}$$

where $A_a$ is the average number of electrons per atom in residue $a$, $r \in \mathbb{R}^3$ and $\sigma = 2$ by default. Hence, the protein's electron density is approximated as the sum of Gaussian kernels centered on its $C_\alpha$ atoms. From these density maps, we then compute an image projection $I \in \mathbb{R}^{N_{\text{pix}} \times N_{\text{pix}}}$ as:

$$I(R, t, S)(r_x, r_y) = g * \int_{\mathbb{R}} V_{RS+t}(r) dr_z, \tag{2}$$

where $(r_x, r_y) \in \mathbb{R}^2$ are the coordinates of a pixel, $r_z \in \mathbb{R}$ is the coordinate along the $z$ axis, $R \in SO(3)$ is a rotation matrix and $t \in \mathbb{R}^3$ is a translation vector. The abuse of notation $RS + t$ means that every atom of $S$ is rotated according to $R$ and then translated according to $t$. The image is finally convolved with the point spread function (PSF) $g$, which in Fourier space is the contrast transfer function (CTF), see Vulović et al. (2013). Note that the integral can be computed exactly for our choice of approximating the density map as a sum of Gaussian kernels, which significantly reduces the computing time.

### 3.2 MAXIMUM LIKELIHOOD WITH VARIATIONAL INFERENCE

To learn a distribution of the different conformations, we hypothesize that the conformation seen in image $I_i$ depends on a latent variable $z_i \in \mathbb{R}^L$, with prior $p(z_i)$. Let $f_\theta(S_0, z)$ be a function which, for a given base structure $S_0$ and latent variable $z$, outputs a new transformed structure $S$. This

function depends on a set of learnable parameters $\theta$. Then, the conditional likelihood of an image $I^\star \in \mathbb{R}^{N_{\text{pix}} \times N_{\text{pix}}}$ with a pose given by a rotation matrix $R$ and a translation vector $t$ is modeled as $p_\theta(I^\star | R, t, S_0, z) = \mathcal{N}(I^\star | I(R, t, f_\theta(S_0, z)), \sigma_{\text{noise}}^2)$, where $\sigma_{\text{noise}}^2$ is the variance of the observation noise. The marginal likelihood is thus given by

$$p_\theta(I^\star | R, t, S_0) = \int p_\theta(I^\star | R, t, S_0, z) p(z) dz. \tag{3}$$

In practice, the pose $(R, t)$ of a given image is unknown. However, following similar works (Zhong et al., 2021b; Li et al., 2023), we suppose that we can estimate $R$ and $t$ to sufficient accuracy using off-the-shelf methods (Scheres, 2012; Punjani et al., 2017).

Directly maximizing the likelihood (3) is infeasible because one needs to marginalize over the latent variable. For this reason, we adopt the VAE framework, conducting variational inference on $p_\theta(z | I^\star) \propto p_\theta(I^\star | z) p(z)$, and simultaneously performing maximum likelihood estimation on the parameters $\theta$.

Let $q_\psi(z | I^\star)$ denote an approximate posterior distribution over the latent variables. We can then maximize the evidence lower-bound (ELBO):

$$\mathcal{L}(\theta, \psi) = \mathbb{E}_{q_\psi}[\log p_\theta(I^\star | z)] - \mathrm{D}_{\text{KL}}(q_\psi(z | I^\star) || p(z)) \tag{4}$$

which lower bounds the log-likelihood $\log p_\theta(I^\star)$. Here $\mathrm{D}_{\text{KL}}$ denotes the Kullback-Leibler (KL) divergence. In this framework $f_\theta$ is called the decoder and $q_\psi(z | I^\star)$ the encoder.

### 3.3 SEGMENT DECOMPOSITION

To handle the often very low SNR encountered in cryo-EM data, we regularize the transformation of the structure produced by the decoder by restricting it to transforming whole segments of the protein. We fix a maximum number of segments $N_{\text{segm}} \in \{1, \ldots, R_{\text{res}}\}$ and we represent the decomposition of the protein by a stochastic matrix $G \in \mathbb{R}^{R_{\text{res}} \times N_{\text{segm}}}$. The rows of $G$ represent "how much of each residue belongs to each segment", and our objective is to ensure that each residue *primarily* belongs to one segment, that is:

$$\forall i \in \{1, \ldots, R_{\text{res}}\}, \exists m^\star \in \{1, \ldots, N_{\text{segm}}\}$$
$$\text{s.t} \sum_{m \neq m^\star} G_{im} \ll 1 \tag{5}$$

We also aim for the segments to respect the sequential structure of the amino acid chain, and the model to be end-to-end differentiable. Without end-to-end differentiability, we could not apply the reparameterization trick and we would have to resort to Monte Carlo estimation of the gradient of the segments, which has a higher variance, see e.g. Mohamed et al. (2019).

To meet these criteria, we fit a Gaussian mixture model (GMM) with $N_{\text{segm}}$ components on the real line supporting the residue indices. Each component $m$ has a mean $\mu_m$, standard deviation $\sigma_m$ and a logit weight $\alpha_m$. The $\{\alpha_m\}$ are passed into a softmax to obtain the weights $\{\pi_m\}$ of the GMM, ensuring they are positive and summing to one. We further anneal the Gaussian components by a temperature $\tau > 0$, and define the probability that a residue $i$ belongs to segment $m$ as:

$$G_{im} := \frac{\{\phi(i | \mu_m, \sigma_m^2) \pi_m\}^\tau}{\sum_{k=1}^{N_{\text{segm}}} \{\phi(i | \mu_k, \sigma_k^2) \pi_k\}^\tau} \tag{6}$$

where $\phi(x | \mu, \sigma^2)$ is the unidimensional Gaussian probability density function with mean $\mu$ and variance $\sigma^2$ and $\tau$ is a fixed hyperparameter. If $\tau$ is sufficiently large, we can expect condition (5) to be verified. See Figure 2 for an example of a segment decomposition using a Gaussian mixture.

In this "soft" decomposition of the protein, each residue can belong to more than one segment, allowing for smooth deformations. In addition, the differentiable architecture is amenable to gradient descent methods, and a well chosen $\tau$ can approximate a "hard" decomposition of the protein. We set $\tau = 20$ in the experiment section. In our experience, this segmentation procedure is very robust to different initialization and converges in only a few epochs.

### 3.4 DECODER ARCHITECTURE

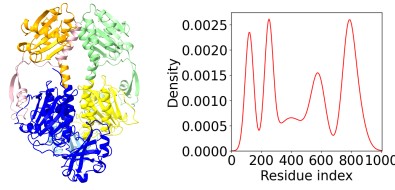

Figure 2: Example of segments recovered with a Gaussian mixture of 6 components.

The decoder describes the distribution of the images given the latent variables, which include:

1. One latent variable $z_i \in \mathbb{R}^L$ per image, parameterizing the conformation.

2. The global parameters $\{\mu_m, \sigma_m, \alpha_m\}_{m=1}^{N_{segm}}$ of the GMM describing the segment decomposition.

Given these latent variables and a base structure $S_0$, we parameterize the decoder $f_\theta$ in three steps. First, a neural network with parameters $\theta$ maps $z_i \in \mathbb{R}^L$ to a set of rigid body transformations, one for each segment $m = 1, \ldots, N_{segm}$. The transformation of segment $m$ is represented by a translation vector $\vec{t}_m$ and a unit quaternion $\vec{q}_m$ (Vicci, 2001), which can further be decomposed into an axis of rotation $\vec{\phi}_m$ and rotation angle $\delta_m$. Second, given the parameters of the GMM, we compute the matrix $G$. Finally, for each residue $i$ of $S_0$, we update the coordinates of all its atoms $\{a_{ik}\}_{k=1}^{A_i}$:

1. First, $a_{ik}$ is successively rotated around the axis $\vec{\phi}_m$ with an angle $G_{im}\delta_m$ for $m \in \{1, \ldots, N_{segm}\}$ to obtain updated coordinates $a'_{ik}$.

2. Second, it is translated according to: $a''_{ik} = a'_{ik} + \sum_{j=m}^{N} G_{im}\vec{t}_m$.

This way, the transformation for a residue incorporate contributions from all segments, proportionally on how much they belong to the segments. If condition (5) is met, a roughly rigid motion for each segment can be expected.

### 3.5 ENCODER AND PRIORS

We follow the classical VAE framework. The distribution $q_\psi(y|I^\star)$ is given by a normal distribution $\mathcal{N}(\mu(I^\star), \text{diag}(\sigma^2(I^\star)))$ where $\mu \in \mathbb{R}^L$ and $\sigma \in \mathbb{R}_+^L$ are generated by a neural network with parameters $\psi$, taking an image $I^\star$ as input. Additionally, the approximate posterior distribution on the parameters of the GMM is chosen to be Gaussian and independent of the input image:

$$\mu_m \sim \mathcal{N}(\nu_{\mu_m}, \beta_{\mu_m}^2)$$
$$\sigma_m \sim \mathcal{N}(\nu_{\sigma_m}, \beta_{\sigma_m}^2)$$
$$\alpha_m \sim \mathcal{N}(\nu_{\alpha_m}, \beta_{\alpha_m}^2)$$

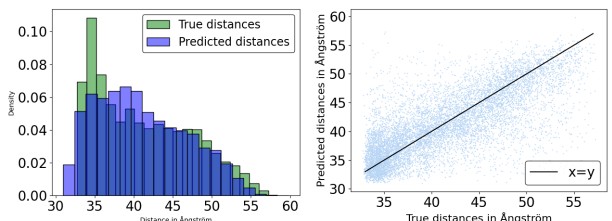

Figure 3: MD dataset SNR 0.001. Left: Histograms of the distances of the two upper domains. The true distances are in green. The recovered distances are in blue. Right: Predicted against true distances in Ångström. The black line represent $x = y$. The correlation between the predicted and true distances is 0.73. For the same plot for cryoStar, see Appendix B.2 of the supplementary file.

where $\{\nu_{\mu_m}, \beta_{\mu_m}, \nu_{\sigma_m}, \beta_{\sigma_m}, \nu_{\alpha_m}, \beta_{\alpha_m}\}_{m=1}^{N_{segm}}$ are parameters that are directly optimized. In practice we use ELU+1 layers for $\sigma_m$ to avoid negative or null standard deviation.

Finally, we assign standard Gaussian priors to both the local latent variable $z_i \sim \mathcal{N}(0, I_L)$, and the global GMM parameters $\{\mu_m, \sigma_m, \alpha_m,\}_{m=1}^{N_{segm}}$. This reparameterization (Kingma & Welling, 2014) is straightforward for a Gaussian distribution. Calculating the KL-divergence between two Gaussian distributions as in equation 4, is also straightforward.

### 3.6 LOSS

Since the images may be preprocessed in unknown ways before running cryoSPHERE, we use a correlation loss between predicted and ground truth image instead of a mean squared error loss,

similar to (Li et al., 2023):

$$\mathcal{L}_{\text{corr}} = \frac{-I_i^\star \cdot I(R_i, t_i, f_\theta(S_0, z))}{||I_i^\star|| \times ||I(R_i, t_i, f_\theta(S_0, z))||} \tag{7}$$

where $\cdot$ denotes the dot product. The total loss to minimize writes:

$$\mathcal{L}(I, I^\star) = \mathcal{L}_{\text{corr}} + \text{D}_{\text{KL}}(q_\psi(z|I^\star)||p(z)) \tag{8}$$

In our experience, it is unnecessary to add any regularization term to the correlation and KL divergence losses, except for datasets featuring a very high degree of heterogeneity. In that case, we offer the option of adding a continuity loss to avoid breaking the protein and a clashing loss to avoid clashing residues, as it is done in (Rosenbaum et al., 2021; Li et al., 2023; Jumper et al., 2021). We describe these losses in Appendix A.1 of the supplementary file.

## 4 RELATED WORKS

Two of the most popular methods for cryo-EM reconstruction, which are *not* based on deep learning, are RELION (Scheres, 2012) and cryoSPARC (Punjani et al., 2017). Both methods perform volume reconstruction, hypothesize that $k$ conformations are present in the dataset and perform maximum a posteriori estimation over the $k$ density maps, thus performing discrete heterogeneous reconstruction. Both of these algorithms operate in Fourier space using an expectation-maximization algorithm Dempster et al. (1977) and are non-amortized: the poses are refined for each image. Other approaches perform continuous heterogeneous reconstruction. For example, 3DVA (Punjani & Fleet, 2021b) uses a probabilistic principal component analysis model to learn a latent space.

Another class of methods involve deep learning and typically performs continuous heterogeneous reconstruction using a VAE architecture. Of those that attempt to reconstruct a density map, cryo-DRGN (Zhong et al., 2020; 2021a) and CryoAI (Levy et al., 2022) use a VAE acting on Fourier space to learn a latent space and a mapping that associates a 3D density map with each latent variable. They perform non-amortized and amortized inference over the poses, respectively. Other methods are defined in the image space, e.g. 3DFlex (Punjani & Fleet, 2021a) and cryoPoseNet (Nashed et al., 2021). They both perform non-amortized inference over the poses. These methods either learn, for a given image $I_i$, $\{V_i(x_k)\}$ the values at a set of $N_{\text{pix}}^3$ fixed 3D coordinates $\{x_k\}$, representing the volume on a grid (*explicit* parameterization), or they learn an actual function $\hat{V}_i : \mathbb{R}^3 \to \mathbb{R}$ in the form of a neural network that can be queried at chosen coordinates (*implicit* parameterization). These volume-based methods cannot use external structural restraints or force fields as additional information. This limits their applicability to low SNR data sets, which are frequent in protein cryo EM.

Other deep learning methods attempt to directly reconstruct structures instead of volumes and share a common process: starting from a plausible base structure, obtained with e.g. AlphaFold (Jumper et al., 2021), for each image, they move each residue of the base structure to fit the conformation present in that specific image. These methods differ on how they parameterize the structure and in the prior they impose on the deformed structure or the motion of the residues. For example AtomVAE (Rosenbaum et al., 2021) considers only residues and penalizes the distances between two subsequent residues that deviate too much from an expected value. CryoFold (Zhong et al., 2021b) considers the residue centers and their side-chain and also imposes a loss on the distances between subsequent residues and the distances between the residue centers and their side-chain. Unfortunately, due to the high level of noise and the fact that we observe only projections of the structures, these "per-residue transformation" methods tend to be stuck in local minima, yielding unrealistic conformations unless the base structure is taken from the distribution of conformations present in the images (Zhong et al., 2021b), limiting their applicability on real datasets. Even though AtomVAE (Rosenbaum et al., 2021) could roughly approximate the distribution of states of the protein, it was not able to recover the conformation given a specific image.

To reduce the bias that the base structure brings, DynaMight (Schwab et al., 2023) fits pseudo-atoms in a consensus map with a neural network directly. Similar to our work, several other methods constrain the atomic model to rigid body motions. For example e2gmm (Chen & Ludtke, 2021; Chen et al., 2023) deform a nominal structure $S_0$ based on how much its residues are close to a

learnt representation $S_{\text{small}}$ of $S_0$. This is similar to our GMM, except that their takes place in $\mathbb{R}^3$ and is not used to perform rigid body motion. Instead, they ask the user to define the segmentation in a later step. This is in contrast to cryoSPHERE, which learns the motion and the segmentation concurrently. Using DynaMight (Schwab et al., 2024), Chen et al. (2024) developed a focused refinement on patches of the GMM representation of the protein. These patches are learnt using $k$-means on the location of residues and do not depend on the different conformations of the data set. This in contrast to cryoSPHERE where the learning of the segments of the protein is tightly linked to the change of conformation. Concurrently to our work, Li et al. (2023) developed cryoStar which learns to translate each residue independently using a variational auto-encoder. They enforce the local rigidity of the motion of the protein by imposing a similarity loss between the base structure and the deformed structure as well as a clash loss. The interested reader can see Donnat et al. (2022) for an in-depth review of deep learning methods for cryo-EM reconstruction.

The reconstruction methods relying on an atomic model, such as cryoStar, DynaMight or cryoSPHERE offer the possibility to the user to provide prior information via this atomic model. They also offer the possibility of deforming the protein according to chemical force fields. This is not the case of the methods performing volume reconstruction without such an atomic model.

## 5 EXPERIMENTS

In this section, we test cryoSPHERE on a set of synthetic[2] and real datasets with varying level of noise and compare the results to cryoDRGN (Zhong et al., 2020) and cryoStar (Li et al., 2023). CryoDRGN is a state-of-the-art method for continuous heterogeneous reconstruction, in which the refinement occurs at the

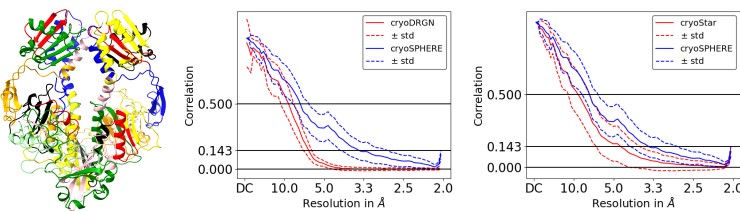

Figure 4: MD dataset. Left: cryoSPHERE Recovered segments. The colors denotes different contiguous domains. Middle and right: mean FSC comparison +/- one standard deviation, for cryoSphere and cryoDRGN and cryoStar. For a comparison between cryoStar and cryoDRGN, see Appendix B.2 in the supplementary file.

level of electron densities, while cryoStar is a structural method similar to ours. To our knowledge, the code for AtomVAE and CryoFold is not available and non-trivial to reimplement. For this reason we focus our comparison on the aforementioned methods, which have furthermore reported state-of-the-art performance. In Appendix B.1, we demonstrate that cryoSPHERE is able to recover the exact ground truth when it exists. We also discuss its performances with varying SNR and $N_{\text{segm}}$ and show how to debias cryoSPHERE results using DRGN-AI or cryoStar volume method in Appendix B.2. Finally, Appendix B.5 compares the computational costs of cryoSPHERE and cryoStar.

### 5.1 MOLECULAR DYNAMICS DATASET: BACTERIAL PHYTOCHROME.

As a more difficult test case we simulate a continuous motion of a bacterial phytochrome, with PDB entry 4Q0J (Burgie et al., 2014). The trajectory starts at the closed conformation of Figure 11 and ends at the most open conformation on the same figure. It corresponds to a dissociation of the two top parts of the protein. This dataset has a very low SNR of $0.001$. Our base structure is obtained by AlphaFold and is subsequently fitted into a homogeneous reconstruction given by the backprojection algorithm. We train cryoSPHERE with $N_{\text{segm}} = 25$, cryoStar, and cryoDRGN for 24 hours each, using the same single GPU. We get one predicted structure per image for cryoSPHERE and cryoStar, that we turn into volumes using (1), and one predicted volume per image for cryoDRGN. See Appendix B.2 in the supplementary file for details and comparison with different values of $N_{\text{segm}}$. Note that since both cryoSPHERE and cryoStar use a nominal structure, we fit the structure we obtained through AlphaFold in the consensus reconstruction obtained by backprojection and use that exact same structure as the nominal one for both methods.

---

[2]See Appendix B of the supplementary file for details on how we created the synthetic datasets.

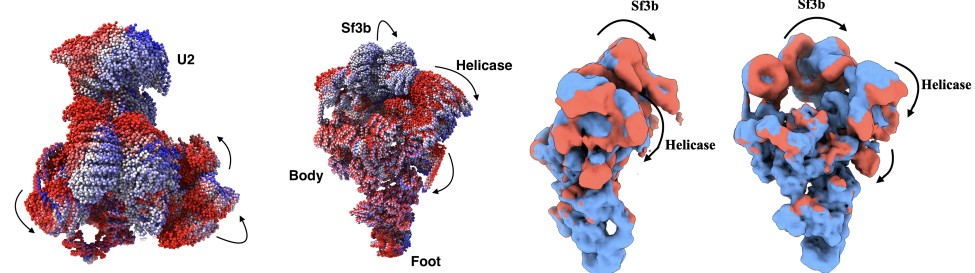

Figure 5: EMPIAR10180. Left and middle left: different views of the structures corresponding to the red dots of Figure 48. The motion goes from red (left in the first principal component) to white to blue (right of the principal component). Only the $C_\alpha$ atoms are shown. Right and middle right: different views of two volumes recovered by training DRGN-AI on the latent space of cryoSPHERE. The U2 domain disappears on the volume because of a compositional heterogeneity.

Figure 3 shows the predicted distance between the two upper parts of the protein being dissociated, against the ground truth distance for each image. In spite of the very low SNR, cryoSPHERE roughly recovers the right distribution of distances. More importantly, the correlation between the predicted distance and ground truth distance is $0.74$, showing that cryoSPHERE is able to recover the correct conformation given an image. This is in stark contrast with Rosenbaum et al. (2021) who could not recover the conformation conditionally on an image. In addition, our model has learnt to separate the two mobile top domains from the fix bottom one, as shown by the segment decomposition in Figure 4. Appendix B.2 in the supplementary file shows the same figures for cryoStar.

We plot the mean of the FSC curves between the predicted volumes and the corresponding ground truth volumes in Figure 4, for cryoSPHERE, cryoDRGN and cryoStar. CryoSPHERE performs better than both cryoDRGN and cryoStar at both the $0.5$ and $0.143$ cutoffs. We attribute this to three key properties. Firstly, we fit our base structure into a consensus reconstruction. This step corrects the position of the medium-scale elements of the base structure that could have been misplaced, boosting the FSC of cryoSPHERE at the $0.5$ cutoff. Secondly, acting directly on the structure level offers a finer resolution than cryoDRGN given the level of noise. Figure 32 shows that cryoDRGN underestimated the opening of the protein and sometimes gives very noisy volumes. That explain why we outperform cryoDRGN at the $0.143$ cutoff. Finally, cryoSPHERE is rigidly moving larger segments of the protein. This provide a better resistance to high levels of noise and overfitting compared to moving each residue individually like cryoStar does, providing a possible explanation to the improvement compared to cryoStar at the $0.143$ cutoff.

## 5.2 EMPIAR 10180

We now demonstrate that cryoSPHERE is applicable to real data as well as large proteins. We run cryoSPHERE on EMPIAR-10180 Plaschka et al. (2017), comprising $327\,490$ images of a pre-catalytic spliceosome with $13\,941$ residues, making it a computationally heavy dataset to tackle. We use the atomic model by Plaschka et al. (2017) (PDB: 5NRL).

Figure 5 shows a set of ten structures taken evenly along the first principal component of the latent space. To interrogate if these structures contain bias from the structural constraints, we perform a volume reconstruction step similar to cryoStar Phase II, see Figure 5.

Traversing the first principal component shows that the Sf3b domain gets incurvated down while the helicase move closer to the foot of the protein. This is in line with the literature (Li et al., 2023; Plaschka et al., 2017). The motion of the protein also brings the alpha helix of the Spp381 domain closer to the foot, as corroborated by Li et al. (2023). Comparison between the recovered structures and volumes (Figure 50) shows similar movements, indicating a small amount of bias from the structural constraints. In addition, the absence of density corresponding to the U2 domain in the volume indicates that it there is compositional heterogeneity that cryoSPHERE could not detect, see Figure 5. We provide a movie of the motion and more structures and volumes in appendix B.3 in the supplementary file.

## 5.3 EMPIAR-12093

We now tackle the recently published EMPIAR-12093 (Bódizs et al., 2024). This dataset comprises two sets of images: one non-activated (Pfr) and one activated (Pr). These dataset are very challeng-

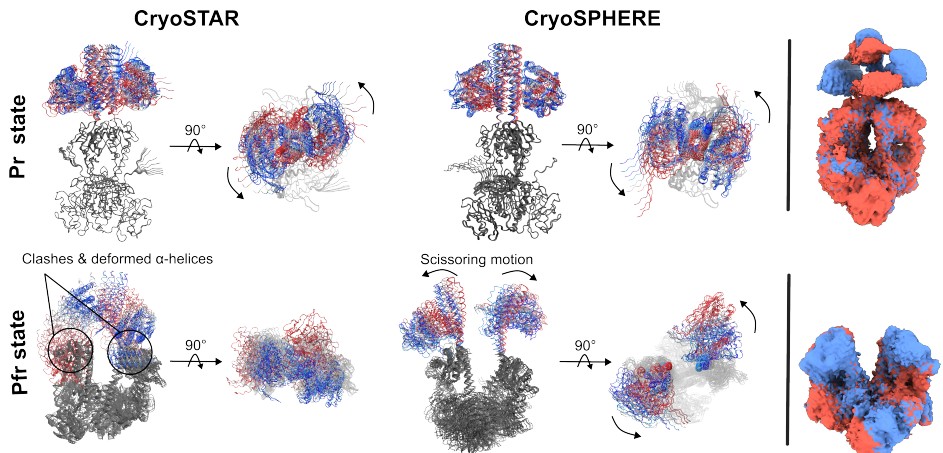

Figure 6: EMPIAR12093. Left of the black line: Ten structures sampled along PC1, for cryoSPHERE and cryoStar. Right of the black line: examples of volumes reconstructed by training the cryoStar volume method on the latent space of cryoSPHERE for debiasing. The blue and red volumes correspond to the first and last volumes along PC1. Top row corresponds to Pr; bottom row to Pfr.

ing because of the high level of noise and heterogeneity of the protein, especially in the Pfr dataset. Traditional methods like cryoSparc (Punjani et al., 2017) or cryoDRGN (Zhong et al., 2020; 2021a) fail at reconstructing the upper part of the protein, see Bódizs et al. (2024) and Appendix B.4.

Figure 6 shows principal component 1 traversal for cryoStar and cryoSPHERE. For Pr, both methods are in strong agreement and reveal a rotation of the upper domain around its axis, while the lower part remains stationary. This aligns with previous studies (Wahlgren et al.; Malla et al., 2024)).

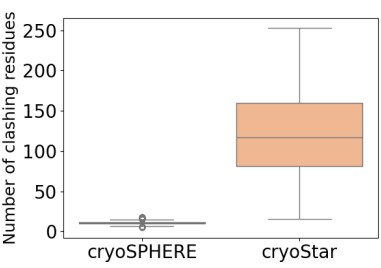

Figure 7: EMPIAR12093. Distribution of the number of clashes for 2000 randomly chosen structures Pfr dataset, for cryoSPHERE and cryoStar. Two non contiguous residues are said to be clashing if their distance is less than 4 Å.

The Pfr dataset showcases an even lower SNR and more dynamical protein: the protein opens up completely. From consensus reconstructions alone, one could suspect that the upper domains are cut off in the sample preparation procedure. However, the protein is complete in Pr (light-activated) structure and the photocycle is reversible,(Takala et al., 2014) suggesting that this is not the case and that strong conformational heterogeneity that is at play.

For Pfr cryoStar is unable to produce physically plausible results: the top part of the protein appears disordered and shows a random motion. In addition, cryoStar does not recover the "scissoring" motion of protein, which is thought to be active (Bódizs et al., 2024). On the contrary, cryoSPHERE gives a high level of motion in a structured manner and recovers the "scissors" opening of the protein. Without any clashes (Fig. 7). (Bódizs et al., 2024).

Analysis of the dataset on phytochromes illustrates the scope and limitations of the different methods. Pure image-based methods (i.e. cryo DRGN) already fail on the Pr state with its intermediate disorder, while cryoSTAR and cryoSPHERE succeed in obtaining reasonable reconstructions (Figure 6). For the Pfr state it becomes evident that cryoSTAR struggles with the high noise and large motions encoded in the dataset. Its deformation-based approach results in unphysical motions along the first principal component, often leading to structural clashes. In contrast, cryoSPHERE handles the noise effectively, producing physically plausible large-scale motions in both the upper and lower domains, see Figure 7, the supplementary movies and B.4. We assign this superior performance to the higher degree of structural constraints that are used in cryoSPHERE compared to cryoSTAR.

We also performed debiasing of cryoSPHERE with a volume method and show examples of reconstructed volumes in Figure 6. For Pr, recovered densities are visible for the entire protein and confirm the dynamics of the upper domains, confirming the absence of compositional heterogeneity and a minimum of bias due to structural constraints. However, for Pfr, meaningful density of the upper (dynamic) part of the protein cannot be recovered, because the signal level in the averaged

density is too low. Thus, for this most dynamic protein case, volume-based debiasing is not possible, despite the fact that the structure based cryoSPHERE finds solutions that fit the data set.

# 6    DISCUSSION

CryoSPHERE presents several advantages compared to other methods for volume and structure reconstruction.

**Efficiency in Deformation:**    Deforming a base structure into a density map avoids the computationally expensive $N_{\text{pix}}^2$ evaluation required by a decoder neural network in methods implicitly parameterising the grid, such as Zhong et al. (2021a); Levy et al. (2022). Furthermore, direct deformation of a structure directly avoids the need for subsequent fitting into the recovered density map.

**Reduced Dimensionality and Noise Resilience:**    Learning one rigid transformation per segment, where the number of segments is much smaller than the number of residues, reduces the dimensionality of the problem. This results in a smaller neural network size compared to approaches acting on each residues, such as Rosenbaum et al. (2021). Rigidly moving large portions of the protein corresponds to low-frequency movements, less prone to noise pollution than the high-frequency movements associated with moving each residue independently. In addition, since our goal is to learn one rotation and one translation per segment, a latent variable of dimension $6 \times N_{\text{segm}}$ is, in principle, a sufficiently flexible choice to model any transformation of the base structure. Choosing the latent dimension is more difficult for volume reconstruction methods such as (Zhong et al., 2021a).

**Interpretability:**    CryoSPHERE outputs segments along with one rotation and one translation per segment, providing valuable and interpretable information. Practitioners can easily interpret how different parts are moving based on the transformations the network outputs. This interpretability is often challenging for deep learning models such as Zhong et al. (2021a); Rosenbaum et al. (2021).

Section 5 and Appendix B.2 demonstrate cryoSPHERE's capability to recover conformational heterogeneity while performing structure reconstruction. The division into $N_{\text{segm}}$ is learned from the data and only marginally impacts the FSC to the ground truth. Moreover, cryoSPHERE recovers the correct motion for the entire range of $N_{\text{segm}}$ values and is able to keep the minimum necessary number of domains when the user sets it too high (Appendix B.1).

**Structural restraints allow interpretation of low SNR datasets:**    It is evident that structural restraints as implemented in cryoSPHERE (this work) and cryoSTAR provide additional restraints that pure volume methods (i.e. cryoDRGN) lack, thus giving better reconstructions for high noise data sets. The additional restraints may introduce bias, which needs to be alleviated using a backprojection algorithm. This, combined with cryoSPHERE's latent space, achieves better $0.5$ cutoffs than cryoDRGN, indicating its effectiveness in resolving conformational heterogeneity and debiasing the results. If such a volume is unavailable, simply increasing $N_{\text{segm}}$ can reduce the bias. As a note of caution we find that for most dynamic protein studies here (the Pfr state of the phytochrome), we find that volume-based debiasing fails because of the very low electron density levels in the reconstructions. Here, other metrics should be developed in the future.

**Summary:**    Our study opens up for significant advancements in predicting protein ensembles and dynamics, critically important for unraveling the complexity of biological systems. By predicting all-atom structures from cryo-EM datasets through more realistic deformations, our work lays the foundation for extracting direct insights into thermodynamic and kinetic properties. This work is an important milestone in showing that one can learn a segmentation of the protein that is intimately linked to the change of conformation of the underlying protein, in an end-to-end fashion. In the future, we anticipate the ability to predict rare and high-energy intermediate states, along with their kinetics, a feat beyond the reach of conventional methods such as molecular dynamics simulations.

It would be interesting to assess how much our segmentation correlates with bottom-up segmentation into domains conducted on the "omics" scale, see e.g. Lau et al. (2023). To achieve this quantitatively, we would need many examples of moving segments from cryo-EM investigations to match the millions of segments from the "omics" studies. Therefore, we leave this investigation to later work.

ACKNOWLEDGEMENT

This work was financially supported by the Wallenberg AI, Autonomous Systems and Software Program (WASP) and the Data-Driven Life Science Program (DDLS) funded by the Knut and Alice Wallenberg Foundation through the WASP-DDLS collaboration, the Swedish Research Council (project no: 2020-04122, 2024-05011), and the Excellence Center at Linköping–Lund in Information Technology (ELLIIT). Our computations were enabled by the Berzelius resource at the National Supercomputer Centre, provided by the Knut and Alice Wallenberg Foundation.

We thank Claudio Mirabello at the National Bioinformatics Infrastructure Sweden at SciLifeLab for providing us access to his AlphaFold installation on Berzelius and Nancy Pomarici for providing input files and explanation of the metadynamics simulation.

## 7 REPRODUCIBILITY STATEMENT

As part of the current paper, we provide a github link to the source code in Section 1. We also describe in detail how we generate the synthetic datasets in Appendix B.2 and the hyperparameters chosen to run cryoStar, cryoSPHERE and cryoDRGN in Appendix B for each of the experiments.

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
