# A METHOD

In this appendix we provide more details on cryoSPHERE.

## A.1 LOSS

In our experience it is not necessary to add any regularization term to the loss Equation 4, except for the datasets featuring a very high level of conformational heterogeneity. In that case, we offer the option to add a continuity loss, which prevents the network from breaking the protein, as well as a clashing loss, which prevents clashing between different residues. The idea of a continuity loss has been introduced in (Jumper et al., 2021) and exploited in cryoEM in (Li et al., 2023; Rosenbaum et al., 2021). The clashing loss has also been introduced in (Jumper et al., 2021) and has been exploited in (Li et al., 2023).

For two subsequent residues belonging to a same chain, we define the continuity loss as:

$$\mathcal{L}_{\mathbf{cont}} = \frac{1}{N_{\mathbf{cont}}} \sum_i^{N_{\mathbf{cont}}} ||d_i - \hat{d}_i||^2 \tag{9}$$

where $N_{\mathbf{cont}}$ is the number of pairs of residues that are subsequent in the entire protein, $d_i$ is the distance between the two residues in pair $i$ in the base structure $\mathcal{S}_0$ and $\hat{d}_i$ is the predicted distance for the corresponding pair. The form of our continuity loss is similar to (Li et al., 2023).

We define the clashing loss as:

$$\mathcal{L}_{\mathbf{clash}} = \frac{1}{N_{\mathbf{clash}}} \sum_i^{N_{\mathbf{clash}}} ||\hat{d}_i - k_{\mathrm{clash}}||^2 \tag{10}$$

where $N_{\mathbf{clash}}$ denotes the number of clashing residues in the protein, where two residues are said to clash if $\hat{d}_i < k_{\mathrm{clash}}$, with $k_{\mathrm{clash}} = 4$ by default. For very large proteins with a high number of residues, computing this clashing loss is impractical. In that case, we compute it for pairs of residues that are distant from 4 to 10Å in the base structure $S_0$. Note that our clashing loss takes into account all of the residues. This is not the case of Li et al. (2023), who implements a similar form as our clashing loss for big proteins while they describe the same loss as we do their paper.

# B EXPERIMENTS

In this appendix, we provide more details on the experiments of Section 5. We followed the same approach to create all the images of the synthetic datasets. We first pose the ground truth structure, which we then convert into a volume, which we then project into a 2D image according to our image formation model in (2) with $\sigma = 2$. After that, we corrupt all the images according to the same CTF parameters described in Table 1. Finally Gaussian noise is added to achieve different SNRs. Here, SNR is defined as the ratio of the variance of the images to the variance of the noise. In this context, the poses are assumed to be exactly known. However, since we use a structure $S_0$, which is different from the structures used to generate the datasets (unless stated otherwise), these poses can only be an approximation. This will not be the case of cryoDRGN, for which these poses will indeed be exact, as this method does not use a base structure. Consequently, in this context, the comparison may introduce a bias in favor of cryoDRGN.

## B.1 TOY DATASET

For this experiment, we predict the phytochrome structure using AlphaFold multimer Evans et al. (2021) on its amino acid sequence with the UniProt The UniProt Consortium (2021) entry Q9RZA4. This protein forms a dimer with 755 residues on each chain. We define two domains for simulation purposes. The first domain comprises the first chain and the first 598 residues of the second chain. The second domain consists of the remaining 157 residues of the second chain. We rotate the second domain around the $(0, 1, 0)$ axis, sampling $10^4$ rotation angles from the Gaussian mixture:

$$0.5 \times \mathcal{N}(-\pi/3, 0.04) + 0.5 \times \mathcal{N}(-2\pi/3, 0.04) \tag{11}$$

| Parameter | Value |
|---|---|
| dfU | 15301.1  Å |
| dfV | 14916.4  Å |
| dfang | 5.28   degrees |
| spherical aberration | 2.7  mm |
| accelerating voltage | 300  keV |
| amplitude contrast ratio | 0.07 |

Table 1: Table of the parameters used to CTF corrupt the generated images. The same values were used for all images of all datasets.

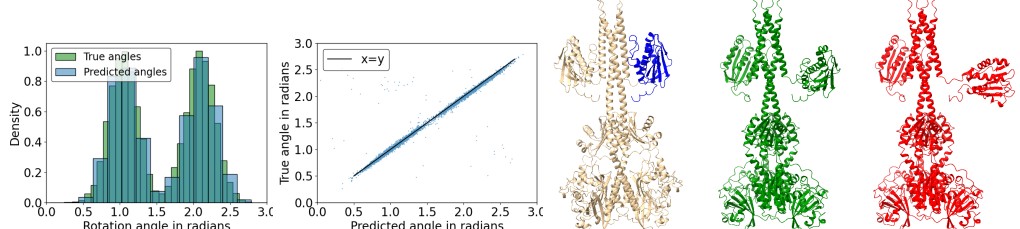

Figure 8: Toy dataset. Leftmost: Histograms of the predicted and true angles of rotation in radians. The true angles are in green. The recovered distances are in blue. Left: Predicted against true angles in Ångström. The black line represent $x = y$. Middle: Base structure with the two domains, also used to generate the images. The domain in blue is rotated according to the axis $(0, 1, 0)$ with angles of rotations sampled from the green distribution on the leftmost figure. Note that the segments predicted by cryoSPHERE exactly match the two domains. The fourth segment is in blue and corresponds to the last 157 residues of chain B, matching exactly the ground truth domain. Right: First mode structure. Rightmost: Second mode structure.

In Figure 8, we present the base structure, the decomposition into the two domains as well as the structures corresponding to the deformed base. These deformations represent the mean rotation of each mode. This gives $10^4$ structures. For each structure, we uniformly sample 15 rotation poses and 15 translation poses on $[-10, 10]^2$. The structure undergoes rotation, translation, and is then turned into an image according to image formation model 2. Subsequently, the images undergo CTF corruption, and noise is added to achieve $\mathrm{SNR} \approx 0.1$. This process generates a total of 150k images, each of size $N_{\mathrm{pix}} = 220$.

We run cryoSPHERE with $N_{\mathrm{segm}} = 4$ for 48 hours on a single NVIDIA A100 GPU, equivalent to 779 epochs. The encoder has 4 hidden layers of size 2048, 1024, 512, 512 and the decoder has two hidden layers of size 350, 350.We use a learning rate of 0.00003 for the parameters of the decoder and encoder and a learning rate of 0.0003 for the segmentation GMM parameters.

Due to computational constraints, the plots in this section are based on only 10000 images, one per conformation.

Testing the segment decomposition, we then run cryoSPHERE by requesting division into $N_{\mathrm{segm}} = 4$. The program learnt a first and third segment with 0 residues, a second segment with 1353 residues and a fourth segment with 157 residues (Figure 8). Thus, cryoSPHERE learnt segments according to the ground truth.

Moreover, Figure 8 shows that most of the predicted angles of rotation for the fourth segment are in excellent agreement with the ground truth structural changes. In addition, the predicted translations for both segments are close to 0, the predicted axis of rotation of the moving segments is close to $(0, 1, 0)$ and the predicted rotation angles for non moving segment are null, see Figure 10.

Finally, Figure 9 illustrates the predicted angles against the latent means, demonstrating that the model effectively learns rotational motion.

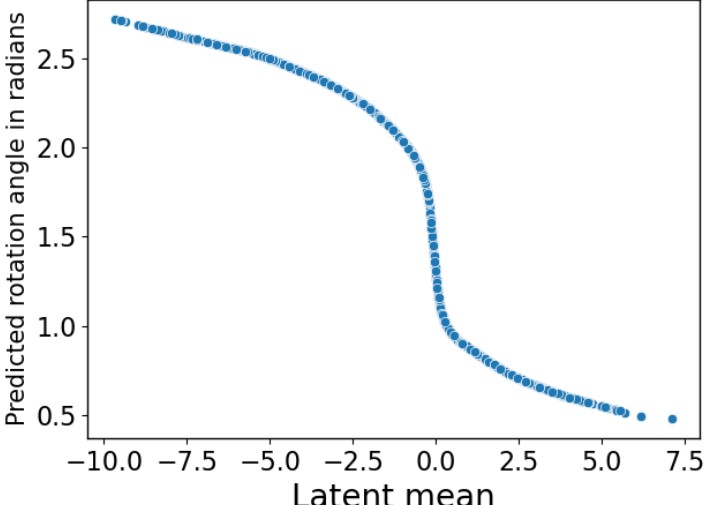

Figure 9: Toy dataset. Predicted angle against latent mean for cryoSPHERE. Note that for clarity 0.3 percent of the points were removed.

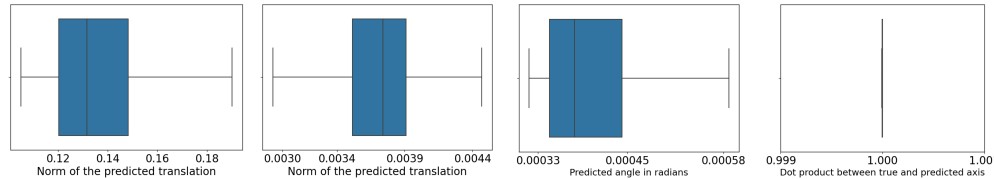

Figure 10: Toy dataset. Leftmost: Boxplot of the norm of the translations predicted for the fourth segment. Middle left: Boxplot of the norm of the translations predicted for the second segment. Middle right: Boxplot of the predicted angle of rotation for the second segment, in radians. Rightmost: Boxplot of the dot product between the predicted axis of rotation for the fourth segment and the true axis of rotation. CryoSPHERE recovers the right axis of rotation almost perfectly.

## B.2 MOLECULAR DYNAMICS DATASET

We take the structure of a phytochrome with PDB ID 4Q0J Burgie et al. (2014) and define two domains: residues 321 to 502 of the first chain and residues 321 to 502 of the second chain. To simulate the dissociation process of the two upper domains, we perform MetadynamicsBarducci et al. (2011) simulations in GROMACS Abraham et al. (2015); Pronk et al. (2013) with the PLUMED 2 implementation Tribello et al. (2014). The collective variable chosen is the distance between the self-defined centers of mass (COMs) of the upper domains (residues 321-502 of chain A and B). A 100 ns simulation is conducted using the NpT ensemble, maintaining pressure control through the Parrinello-Rahman barostat. Gaussian deposition occurs every 5000 steps, featuring a height of 0.1 kJ/mol and a width of 0.05 nm. Afterwards, we extract $10^4$ structures along the dry trajectory. See Figure 11 for examples of structures. The closed conformation is the starting conformation of the MD simulation and the most open one corresponds to the end.

For each structure, we sample 15 rotation poses uniformly together with translations uniformly on $[-10, 10]^2$. This results in a dataset of 150k images. We use $N_{\text{pix}} = 190$, and each pixel is of size 1Å. We finally add Gaussian noise to create three datasets: a SNR $= 0.1$, a SNR $= 0.01$ and SNR $= 0.001$ dataset. The results for SNR $= 0.001$ are described in Section 5.1 and this appendix describes the results for the SNR $= 0.1$ and SNR $= 0.01$ datasets.

For cryoSPHERE on all three datasets, the encoder is a 4-hidden-layer neural network with fully connected hidden layers of dimension of $512, 256, 64, 64$. The decoder is a 2-hidden-layer neural

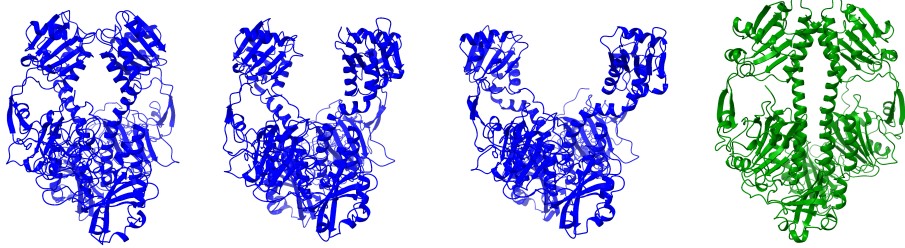

Figure 11: From left to right. 1/ The starting close conformation of the MD simulation. 2/ A medium-open structure arising from the MD simulation. 3/ An open conformation towards the end of the MD simulation. 4/ The AlphaFold structure used as a base structure.

network with fully connected hidden layers of dimension $512, 512$. We set the batch size to $128$ and use the ADAM optimizer Kingma & Ba (2017) with a learning rate of $0.00003$ for the encoder and decoder parameters, while we set a learning rate of $0.0003$ for the GMM segmentation parameters. The latent dimension is set to $8$ and we run the program with $N_{\text{segm}} = 10$, $N_{\text{segm}} = 20$ and $N_{\text{segm}} = 25$ for all datasets. We train for 24 hours on a single NVIDIA A100 GPU.

We use the default parameters for both cryoStar and cryoDRGN but disable the structural loss of cryoStar, except for the elastic network loss. We train both methods for 24 hours on the same GPU as cryoSPHERE. To maintain consistency when comparing volumes generated from cryoDRGN, our methods, cryoStar and the ground truth, we convert the structures (ground truth, predicted with cryoStar and predicted with cryoSPHERE) into volumes using the same image formation model employed to generate the dataset, see 1. Since cryoStar also proposes a volume method similar to cryoDRGN, after training the structure method of cryoStar for 24 hours, we train their volume methods for the same amount of time. We report the results for this volume method in this appendix.

Since cryoSPHERE and cryoStar are structural methods, we can compute the predicted distance between the two domains defined earlier for each image, both for cryoStar and cryoSPHERE.

For computational efficiency, the FSC plots and distances plots are not based on all structures: only one image per structure is used to compute the distances. Consequently, the distance plots are based on 10k images. For the computation of the FSC curves, we select only 1000 images evenly distributed among these 10000 structures.

### B.2.1    SNR $0.1$

This subsection describes the results for cryoSPHERE, cryoStar and cryoDRGN on our molecular dynamics simulation dataset with SNR $0.1$. We use $N_{\text{segm}} = 25$ in this section.

Figures 13 and 14 show that both cryoSPHERE and cryoStar recover the ground truth distribution of distances very well. They are also able to identify the correct conformation conditionally on an image, as illustrated by the predicted versus true distances plot.

Figure 15, shows that both cryoStar and cryoSPHERE outperform cryoDRGN at both the $0.5$ and $0.143$ cutoffs. It seems cryoStar slightly outperforms cryoSPHERE at both cutoffs. This might be because the SNR is rather high, hence moving each residue individually offers a greater flexibility than moving segments, while the risk of overfitting is low. In addition, we can see that the volume method of cryoStar perform similarly, if not worse, than cryoDRGN. That seems to indicate that this volume method does not benefit from the information gained by the structural method. Figure 12 shows three examples of volumes predicted by the volume method of cryoStar, together with the corresponding ground truth.

Figure 16 and 17 show a set of predicted structures compared to the ground truth for cryoSPHERE and cryoStar. Both methods are able to recover the ground truth almost perfectly.

Figure 18 shows examples of cryoDRGN predicted volume together with the corresponding ground truth. The method is able to recover the ground truth volumes almost perfectly.

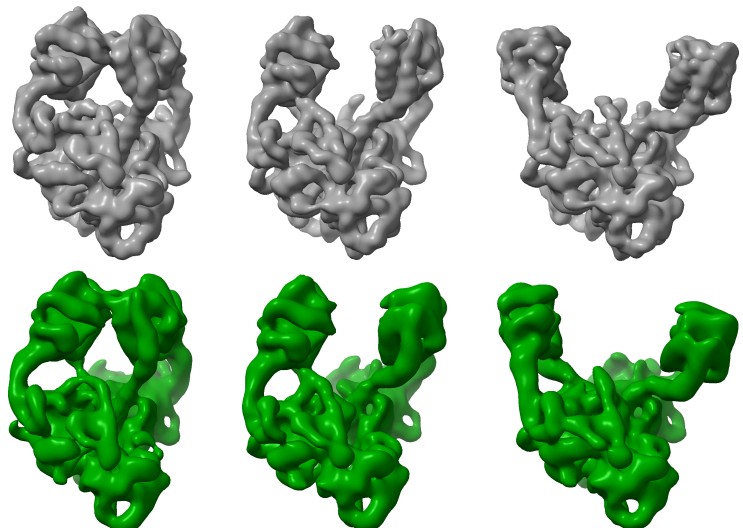

Figure 12: SNR $0.1$. Example volumes produced by the volume method of cryoStar with the corresponding ground truth. Top: ground truth. Bottom: corresponding volumes predicted by the volume method of cryoStar.

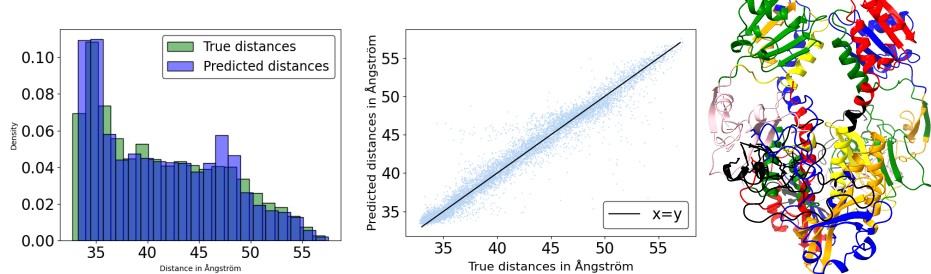

Figure 13: Results for cryoSPHERE on SNR $0.1$ with $N_{\text{segm}} = 25$. Left: distribution of distances predicted by cryoSPHERE compared to the ground truth distribution. Middle: true versus predicted distances for cryoSPHERE. Right: segments decomposition.

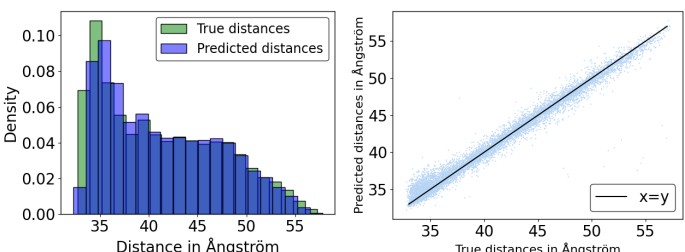

Figure 14: Results for cryoStar on SNR $0.1$. Left: distribution of distances predicted by cryoStar compared to the ground truth distribution. Right: true versus predicted distances for cryoStar

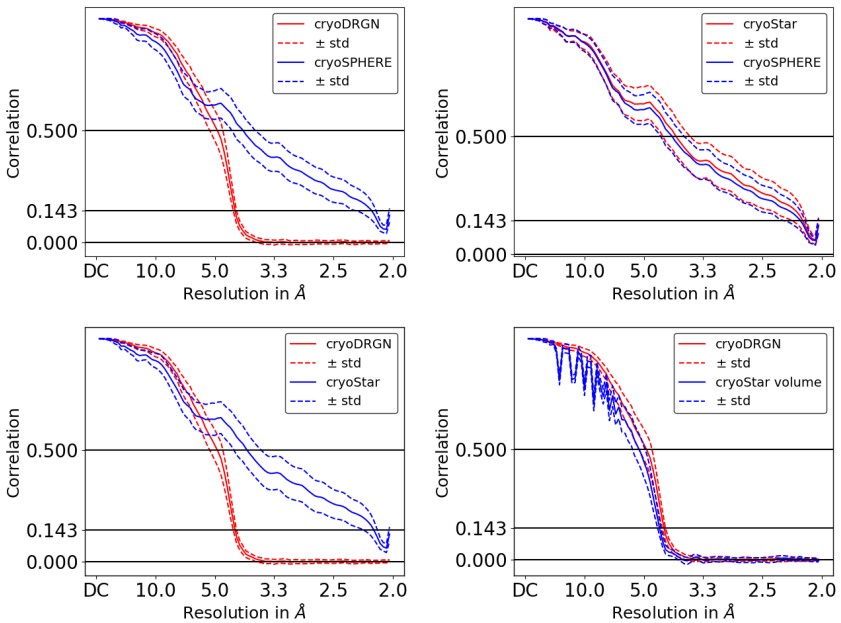

Figure 15: SNR $0.1$. Mean Fourier shell correlation ($\pm$ std) comparison for cryoSPHERE with $N_{\text{segm}} = 25$, cryoStar and cryoDRGN. Top left: cryoSPHERE versus cryoDRGN. Top right: cryoSPHERE vs cryoStar. Bottom left: cryoDRGN vs cryoStar. Bottom right: cryoStar volume method vs cryoDRGN

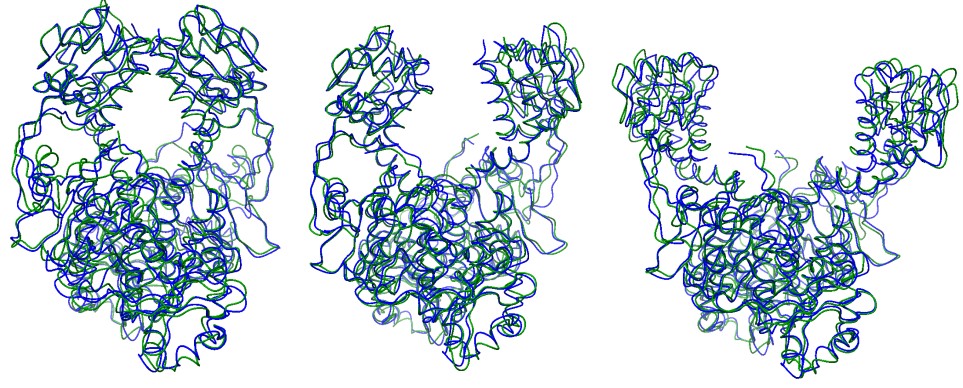

Figure 16: SNR $0.1$. Examples of reconstructed structures by cryoSHERE. Blue is predicted, green is ground truth. CryoSPHERE is able to recover the right conformation. Left to right: image number 11, 5001 and 9999.

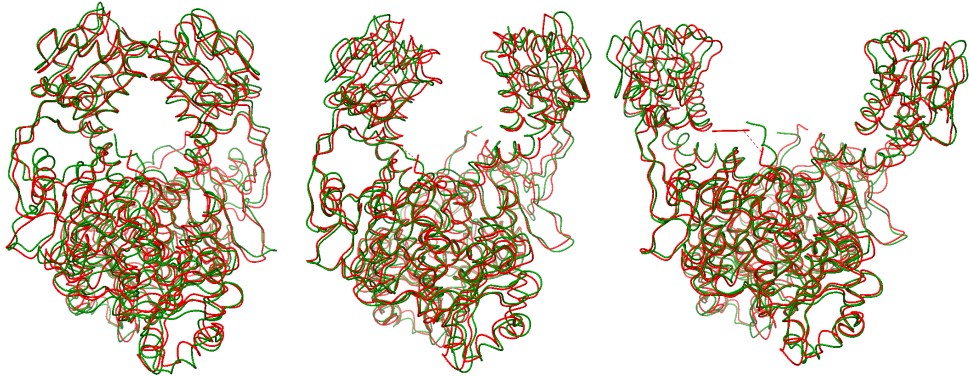

Figure 17: SNR 0.1. Examples of reconstructed structures by cryoStar. Red is predicted, green is ground truth. cryoStar is able to recover the right conformation. Left to right: image number 11, 5001 and 9999.

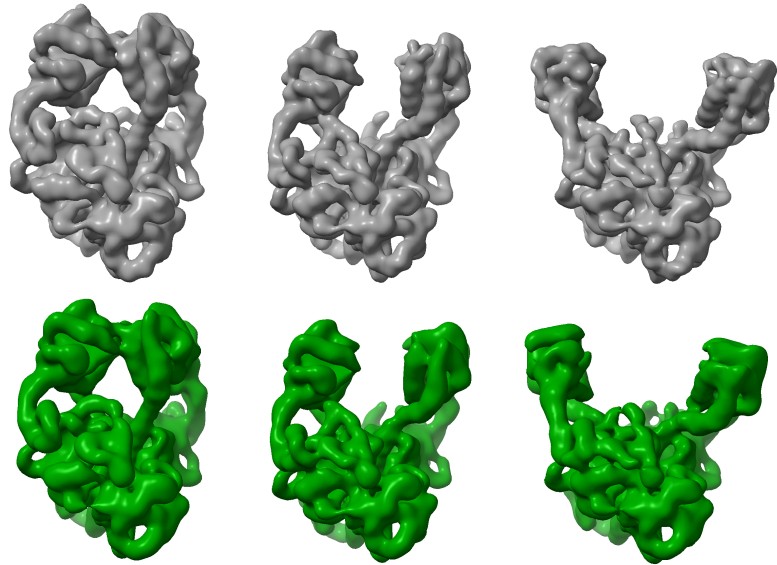

Figure 18: SNR 0.1. Examples of reconstructed volumes by cryoDRGN. Green is cryoDRGN and gray is the corresponding ground truth volume.

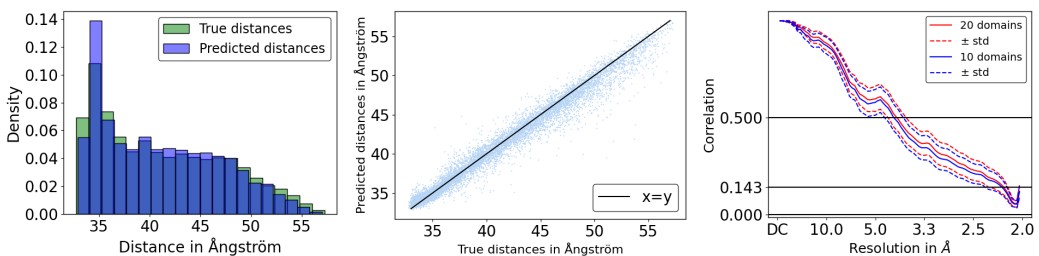

Figure 19: Results for cryoSPHERE on SNR 0.1 with $N_{\text{segm}} = 10$. Left: distribution of distances predicted by cryoSPHERE compared to the ground truth distribution. Middle: true versus predicted distances for cryoSPHERE. Right: FSC comparison between cryoSPHERE with $N_{\text{segm}} = 20$ and $N_{\text{segm}} = 10$.

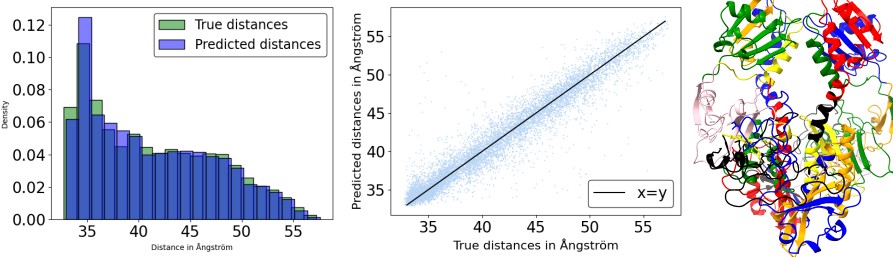

Figure 20: Results for cryoSPHERE on SNR $0.01$ with $N_{\text{segm}} = 25$. Left: distribution of distances predicted by cryoSPHERE compared to the ground truth distribution. Middle: true versus predicted distances for cryoSPHERE. Right: segments decomposition.

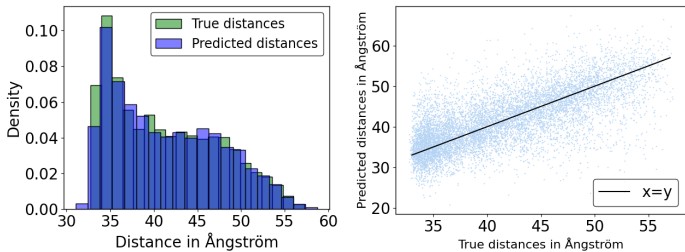

Figure 21: Results for cryoStar on SNR $0.01$. Left: distribution of distances predicted by cryoStar compared to the ground truth distribution. Right: true versus predicted distances for cryoStar

### B.2.2 SNR $0.01$

This subsection describes the results for cryoSPHERE, cryoStar and cryoDRGN on our molecular dynamics simulation dataset with SNR $0.01$.

Figures 20 and 21 show that both cryoSPHERE and cryoStar are able to recover the ground truth distribution of distance. Also, given an image, both methods are able to recover the correct conformation.

Figure 22 shows that both cryoSPHERE and cryoStar are outperforming cryoDRGN at the $0.5$ and $0.143$ cutoffs while having very similar performances. In addition, we can see that the volume method of cryoStar perform similarly, if not worse, than cryoDRGN. That seems to indicate that this volume method does not benefit from the information gained by the structural method. Figure 23 shows three examples of volumes reconstructed with the volume method of cryoStar, together with the corresponding ground truth.

Figure 24 and 25 shows a set of predicted structures compared to the ground truth. Both cryoSPHERE and cryoStar are able to recover the ground truth almost perfectly.

Finally, Figure 26 shows examples of cryoDRGN predicted volume together with the corresponding ground truth. The method is able to recover the ground truth volumes almost perfectly, with a somewhat lower resolution compared to Figure 18.

### B.2.3 SNR $0.001$

This subsection complements the results of Section 5.1.

Figure 27 shows that cryoStar is also able to recover the rough distribution of conformations as well as the correct conformation given an image. The comparisons of FSC shows that cryoStar perform similarly to cryoDRGN at the $0.5$ cutoff but better at the $0.143$. In addition, Figure 28 shows that cryoStar volume method does not perform significantly better than cryoDRGN, in spite of using the information given by the structural method of cryoStar. Figure 31 shows a set of example volumes reconstructed by the volume method of cryoStar and the corresponding ground truth.

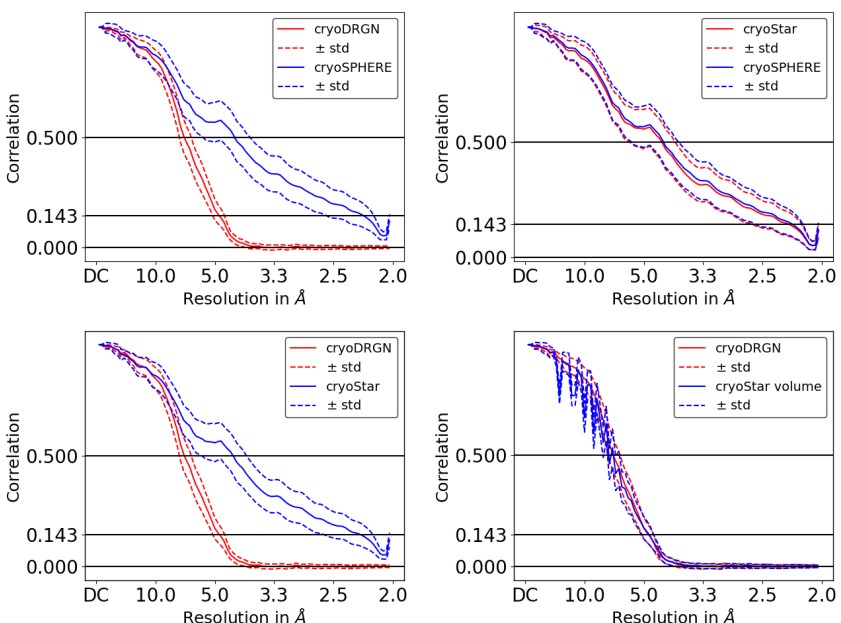

Figure 22: SNR 0.01. Mean Fourier shell correlation (± std) comparison for cryoSPHERE with $N_{segm} = 25$, cryoStar and cryoDRGN. Top left: cryoSPHERE versus cryoDRGN. Top right: cryoSPHERE vs cryoStar. Bottom left: cryoDRGN vs cryoStar. Bottom right: cryoStar volume method vs cryoDRGN.

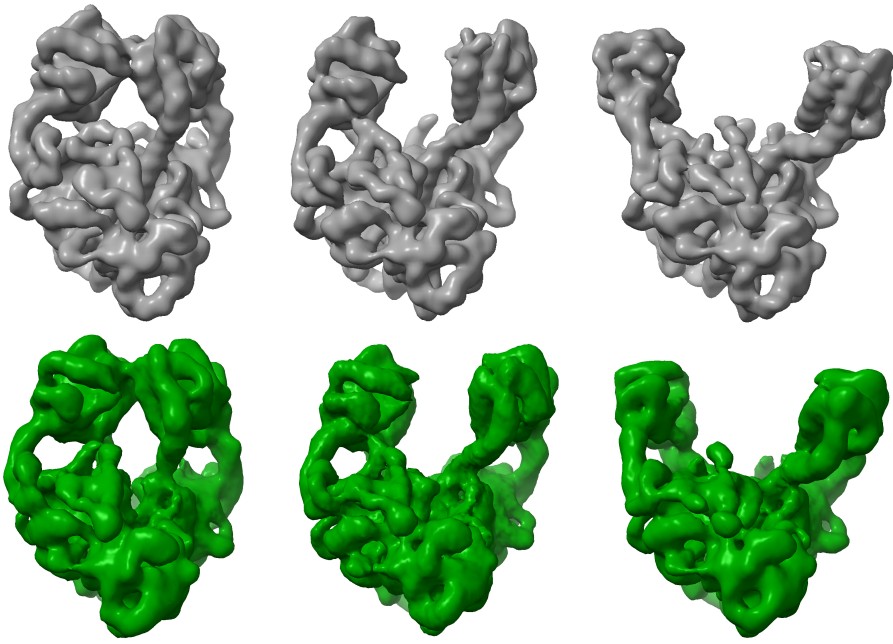

Figure 23: SNR 0.01. Example volumes produced by the volume method of cryoStar with the corresponding ground truth. Top: ground truth. Bottom: corresponding volumes predicted by the volume method of cryoStar.

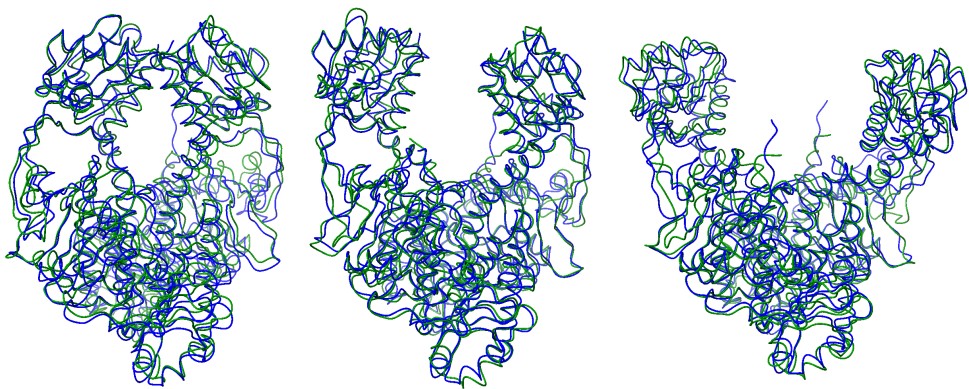

Figure 24: SNR 0.01. Examples of reconstructed structures by cryoSHERE. Blue is predicted, green is ground truth. cryoSPHERE is able to recover the right conformation. Left to right: image number 11, 5001 and 9999.

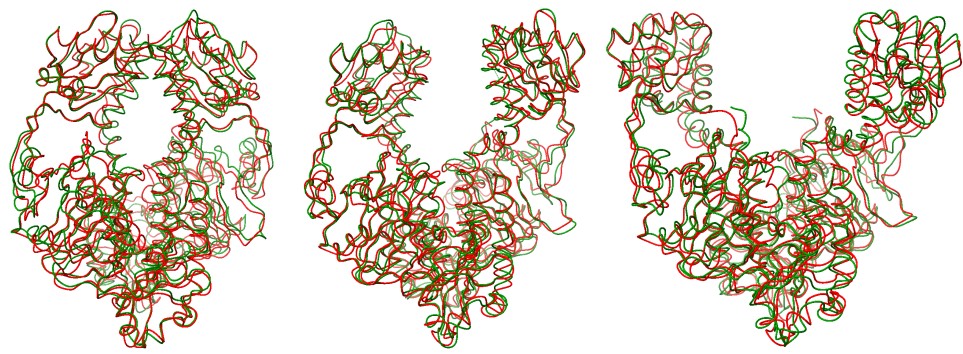

Figure 25: SNR 0.01. Examples of reconstructed structures by cryoStar. Red is predicted, green is ground truth. cryoStar is able to recover the right conformation. Left to right: image number 11, 5001 and 9999.

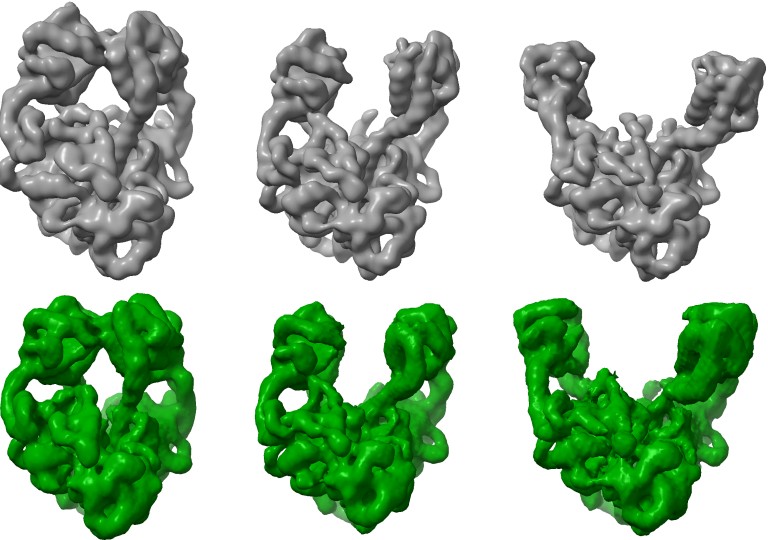

Figure 26: SNR 0.01. Examples of reconstructed volumes by cryoDRGN. Green is cryoDRGN and gray is the corresponding ground truth volume.

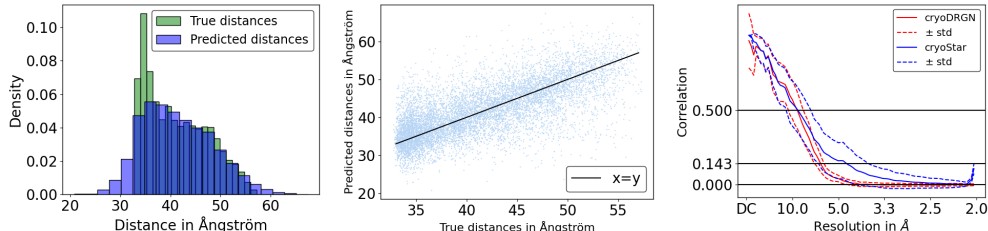

Figure 27: Results for cryoStar on SNR 0.001. Left: distribution of distances predicted by cryoStar compared to the ground truth distribution. Middle: true vs predicted distances for cryoStar. Right: FSC comparison between cryoStar and cryoDRGN

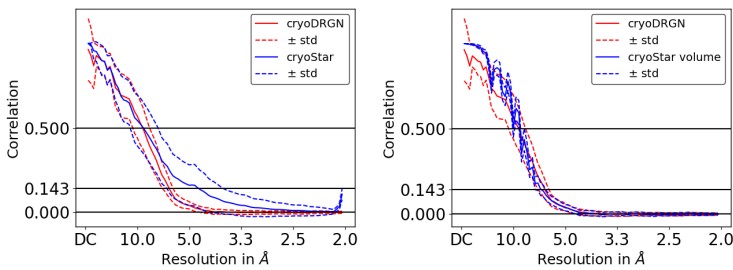

Figure 28: SNR 0.001. Mean Fourier shell correlation ($\pm$ std) comparison for cryoStar and cryoDRGN. Left: cryoStar versus cryoDRGN. Right: cryoStar volume method vs cryoDRGN.

Figure 29 and 30 show a set of predicted structures compared to the ground truth. Both cryoSPHERE and cryoStar are able to recover the ground truth, but in a more approximate fashion than for the higher SNR datasets.

Figure 32 shows examples of cryoDRGN predicted volume together with the corresponding ground truth. The method underestimates the opening of the protein and predicts low resolution volumes with a lot of noise. It seems it is overfitting.

### B.2.4 DEBIASING CRYOSPHERE

When deforming an atomic model to recover different conformations, one should be careful not to bias the results. CryoStar (Li et al., 2023) developed a volume method to help assess the bias

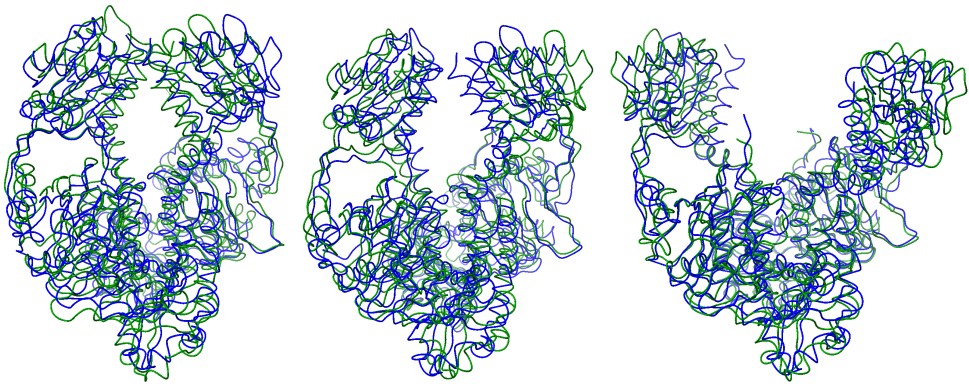

Figure 29: SNR 0.001. Examples of reconstructed structures by cryoSHERE. Blue is predicted, green is ground truth. CryoSPHERE is able to recover the right conformation. Left to right: image number 11, 5001 and 9999.

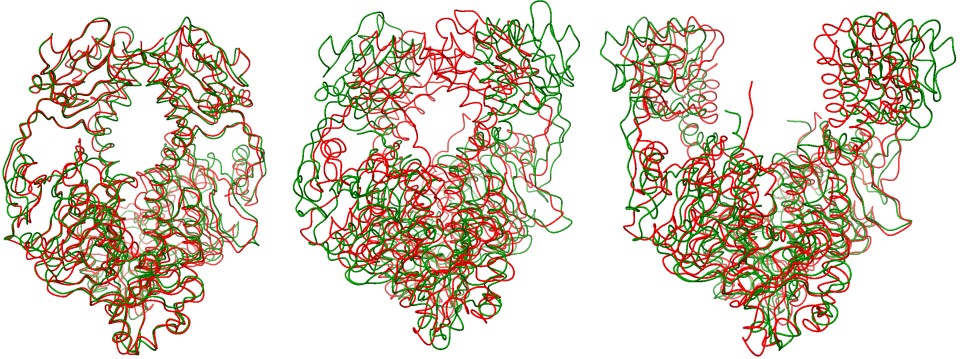

Figure 30: SNR 0.001. Examples of reconstructed structures by CryoStar. Red is predicted, green is ground truth. CryoStar is able to recover the right conformation. Left to right: image number 11, 5001 and 9999.

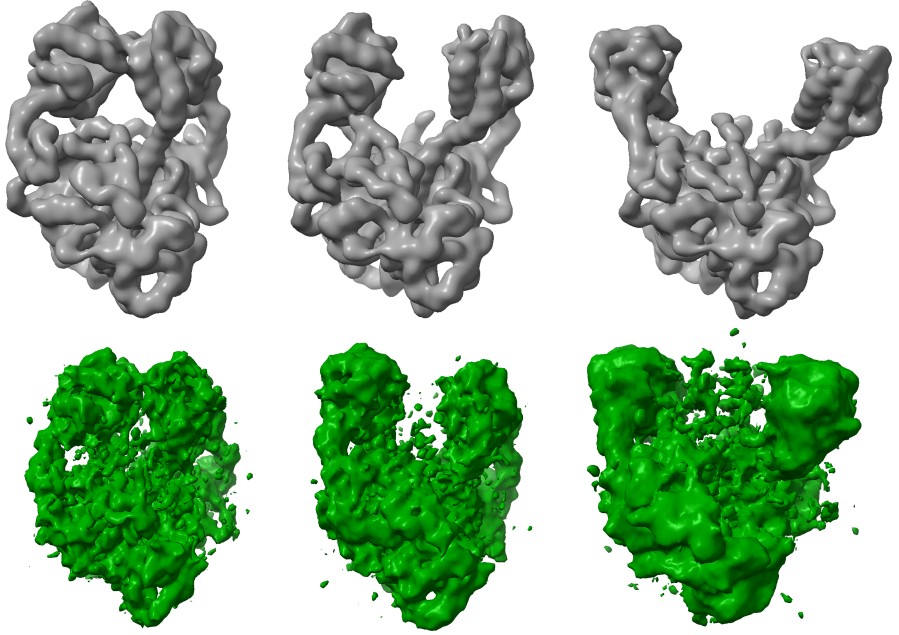

Figure 31: SNR 0.001. Example volumes produced by the volume method of cryoStar with the corresponding ground truth. Top: ground truth. Bottom: corresponding volumes predicted by the volume method of cryoStar.

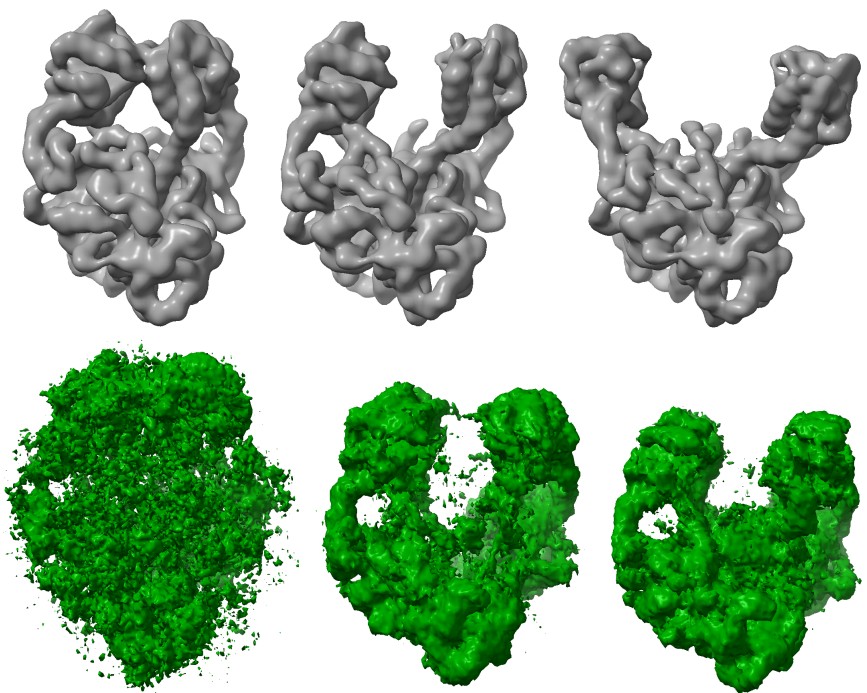

Figure 32: SNR 0.001. Examples of reconstructed volumes by cryoDRGN. Green is cryoDRGN and gray is the corresponding ground truth volume.

induced by the atomic model. However, we demonstrate here that this two stages of training are nothing specific to cryoStar and can, in fact, be applied to any structural method just by using DRGN-AI Levy et al. (2024).

We remove the last 40 residues of chain B of the 10 000 structures obtained through molecular dynamics simulations, see Appendix B.2. We then follow the exact same process to generate 150k images with SNR 0.01. Again, see Appendix B.2.

We run cryoSPHERE with $N_{segm} = 20$ for 24 hours using the same base structure as the other experiments of Appendix B.2, that we obtained through AlphaFold. In other words, our base structure has 40 more residues than the structures on the images, see Figure 33.

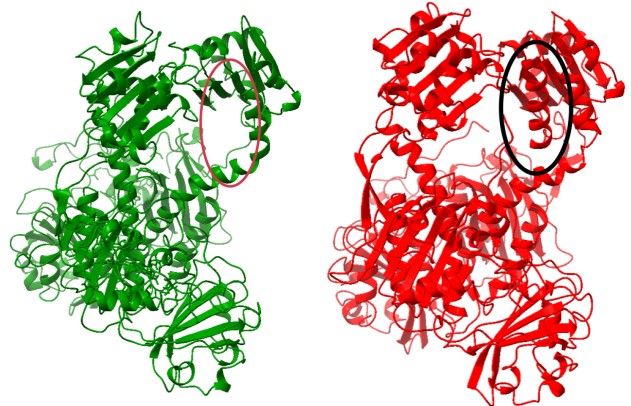

Figure 33: Left: example of a structure used to to generate the images. The missing part is highlighted in a red ellipsoid. Right: the base structure used, with the residues that were removed for the image generation highlighted in a black ellipsoid.

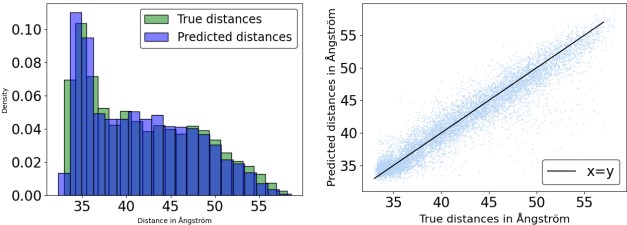

Figure 34: Debiasing the atomic model. Left: Distribution of ground truth and predicted distances. Right: True versus predicted distances.

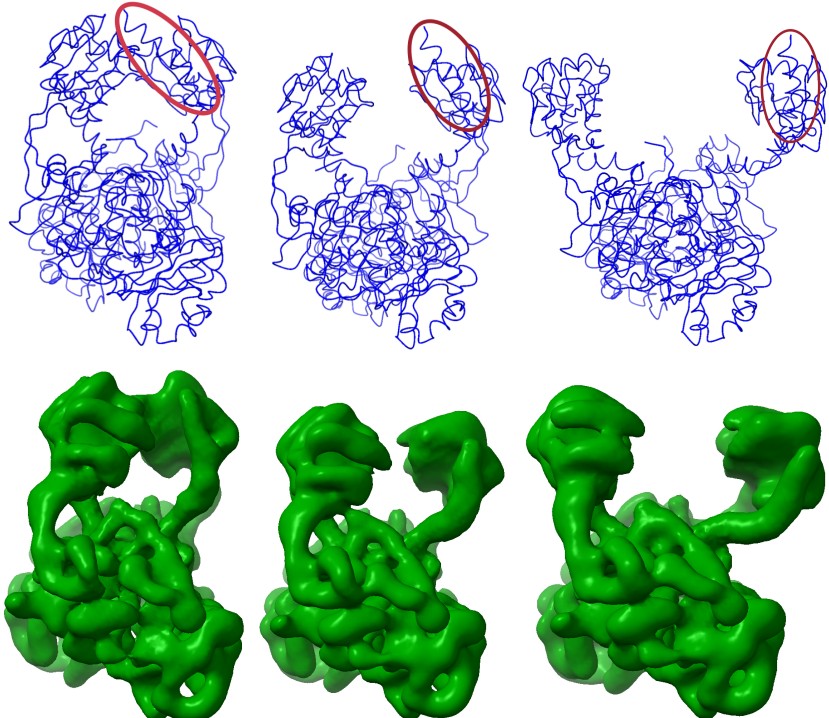

Figure 35: Debiasing the atomic model. Top, from left to right: example of 3 structures predicted by cryoSPHERE. The red ellipses show the alpha helix that is present in the base structure but is not on the images. Bottom, from left to right: corresponding volumes reconstructed by DRGN-AI.

This discrepancy does not affect the predicted distances by cryoSPHERE, as shown in Figure 34. Figure 35 shows 3 structures predicted by cryoSPHERE. Even though the opening motion is correctly recovered, the algorithm tries to remove the missing alpha helix from where there should not be any density. In a sense, cryoSPHERE detects that the atomic model has too many residues, but it cannot temove them, by design. We leave to future work the ability to learn the amplitude of each Gaussian mode of Equation 1. Here, we propose to debias the atomic model by running, in a second step, DRGN-AI with the final latent space of the cryoSPHERE run and fixed poses, similar to what cryoStar proposes. Figure 35 shows the reconstructed volumes by DRGN-AI corresponding to the plotted structures predicted by cryoSPHERE. The alpha helix is not present, which permits to detect the bias brought by the base structure for cryoSPHERE.

In terms of computational cost, on a dataset of 150k images of size $190 \times 190$, DRGN-AI performs one epoch in 10 minutes while the volume method of cryoStar perform one epoch in 6 minutes.

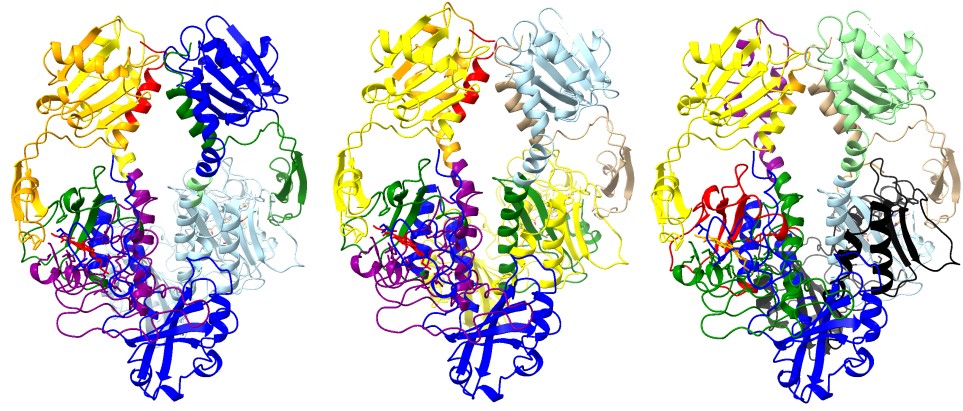

Figure 36: Segmentation of cryoSphere with $N_{\text{segm}} = 10$. Left: SNR 0.1. Middle: SNR 0.01. Right: SNR 0.001.

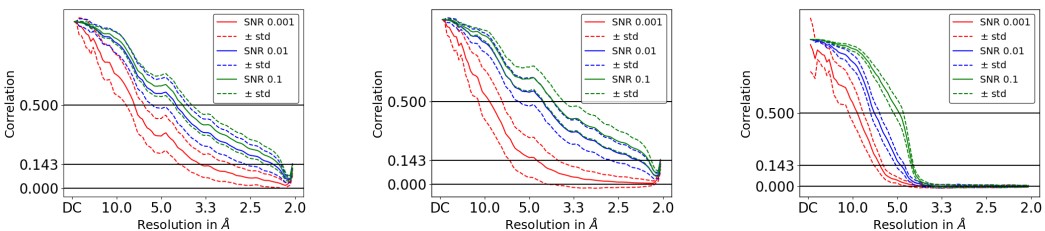

Figure 37: Comparison, for each method, of the Fourier shell correlation for different SNR. Left to right: cryoSPHERE, cryoStar, cryoDRGN.

### B.2.5 Comparison accross SNR for cryoSPHERE, cryoDRGN and cryoStar.

In this section, we look at the change of performance for each method accross the different SNR.

Figure 37 shows that all three methods experience a drop in FSC with decreasing SNR. CryoSPHERE and cryoStar show a drop of one standard deviation between SNR 0.1 and 0.01 while cryoStar experience a much bigger drop than cryoSPHERE between SNR 0.01 and 0.001, which confirms that cryoSPHERE is more resilient to a high level of noise.

CryoDRGN shows a steady decrease in its FSC with decreasing SNR.

The fact that for a SNR of 0.1 cryoStar is slightly outperforming cryoSPHERE (see Appendix B.2.1), that for a SNR of 0.01 cryoSPHERE outperforms cryoStar (see Appendix B.2.2), and that for a SNR of 0.001 cryoSPHERE outperforms cryoStar by one standard deviation (see Section 5.1) confirms that moving big chunks of the protein as rigid bodies is more resilient to low SNR than moving each residue individually.

### B.2.6 Comparison for different values of $N_{\text{SEGM}}$.

In this subsection, we compare the results of cryoSPHERE for three different values of $N_{\text{segm}} = 10, 20, 25$, for the different SNR of the MD dataset in 5.1.

Figure 38 shows the evolution of the FSC curves with different values of $N_{\text{segm}}$ for different SNR. As we can expect, the high the number of segments, the more flexible cryoSPHERE and the better FSC. The lower the SNR, the greater we gain in FSC by increasing $N_{\text{segm}}$. This is because with decreasing SNR, the initial fitting of the structure in a consensus reconstruction is less accurate. Hence, the method benefits from a greater flexibility to adjust the protein on a smaller scale.

Figures 39, 40 41,42,43,44 show the distributions of the predicted and true distances and the true versus the predicted distances for each value of SNR and $N_{\text{segm}}$. This shows that the choice of $N_{\text{segm}}$

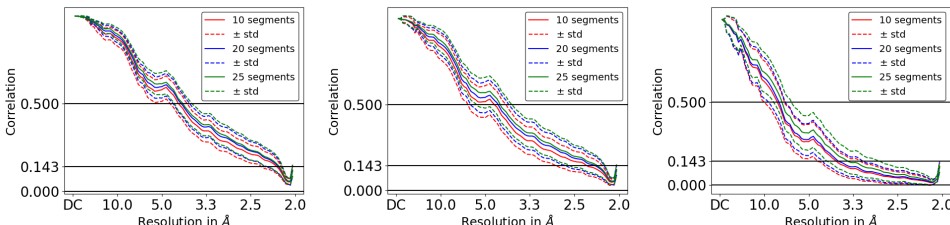

Figure 38: FSC curves for different $N_{segm}$ values. From left to right: SNR 0.1, SNR 0.01, SNR 0.001.

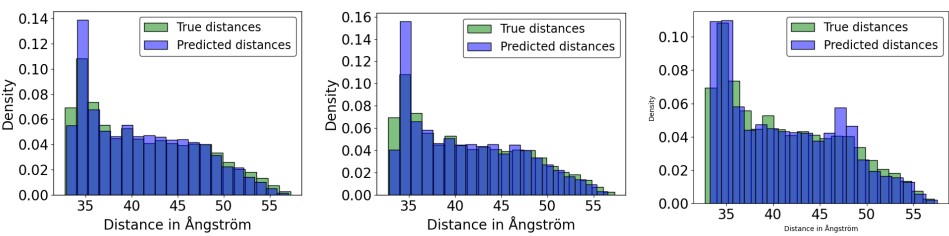

Figure 39: MD dataset, SNR 0.1. Distribution of the predicted distances. From left to right: 10 segments, 20 segments, 25 segments.

is not so critical for cryoSPHERE to work well. The higher $N_{segm}$, the better it is in terms of FSC, but a value of 10 still gives a good performance.

Finally, Figures 45,46and 47 show the segment decomposition for different values of $N_{segm}$. CryoSPHERE always learns a segmentation differentiating the two chains and the top and bottom parts of the protein.

### B.3 EMPIAR-10180

This section gives more details on the experiment with the EMPIAR-10180 datasets described in Section 5.2. The data was processed by Relion, hence the poses and CTF are assumed to be known. We use an encoder with 4 hidden layers of size 512, 256, 64, 64 and a decoder with 2 hidden layers with size 512, 512.

We train cryoSPHERE with no clashing nor continuity loss, with $N_{segm} = 20$. We use the ADAM optimizer with a learning rate of 0.00003 for the parameters of the encoder and decoder, while we set the learning rate to 0.0003 for the parameters of the GMM segmentation.

We low pass filter the images with a bandwith of 23.4Å, we apply a mask of radius 0.9375 to the input images and we apply a mask of radius 1 to the true and predicted images for the computation of the correlation loss.

Figure 49 shows 4 structures taken from the principal component traversal depicted in Figure 48. The structures contain only the $C_\alpha$ atoms. We provide a movie of the traversals of principal component 1 and principal component 2 by clicking here.

We subsequently train DRGN-AI on the latent space provided by cryoSPHERE, similar to cryoStar Phase II of training. We show volumes taken along the first principal component in Figure 50. We recover the correct bending of the protein toward its "foot". In addition, this second step detects the compositional heterogeneity and the density is zero in this region. This is a detail that the structural method of cryoSPHERE could not detect. We successfully identified a bias in the base structure.

We provide a movie of the motion recovered by DRGN-AI trained on the latent variable of cryoSPHERE here.

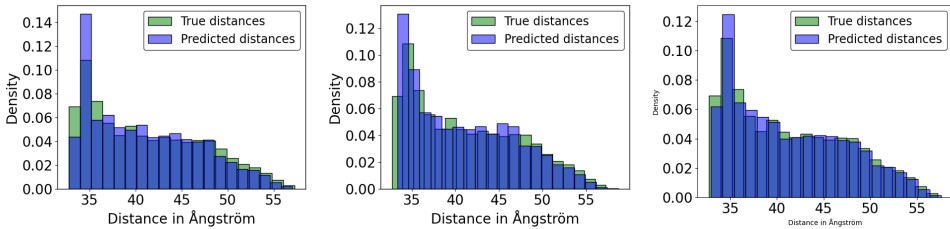

Figure 40: MD dataset, SNR 0.01. Distribution of the predicted distances. From left to right: 10 segments, 20 segments, 25 segments.

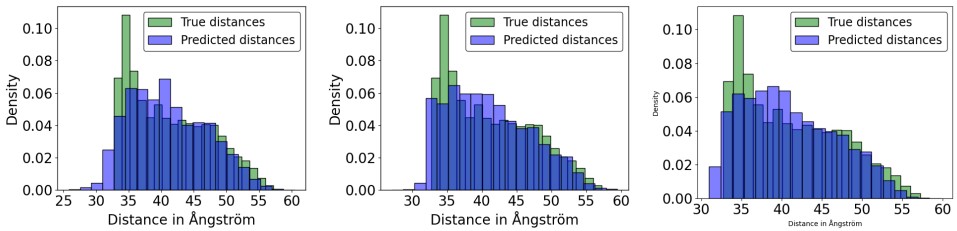

Figure 41: MD dataset, SNR 0.001. Distribution of the predicted distances. From left to right: 10 segments, 20 segments, 25 segments.

### B.4 EMPIAR-12093

This experiment demonstrates that cryoSPHERE is applicable to real data with high noise levels. We applied cryoDRGN, cryoSTAR and cryoSPHERE to a bacterial phytochrome (Bódizs et al., 2024) dataset (medium-sized protein, 120 kDa). This dataset comprises two distinct subsets of 200 000 images of size 400 x 400 each, representing the protein in its light-activated state (red-absorbing state, called Pr) and its resting state (far-red-absorbing state, called Pfr). The pre-processing steps are detailed in Bódizs et al. (2024). We downgrade the images to a size of 256 x 256. We give a computational budget of 24 hours on the same single GPU to cryoDRGN, cryoStar structural method and cryoSphere and compare their results. We subsequently run cryoStar volume method on the latent space obtained by cryoStar and cryoStar volume method on the latent space obtained by cryoSPHERE.

We provide PC 1 traversal movies for both cryoSPHERE and cryoStar structures and volumes for both Pr and Pfr here.

#### B.4.1 PFR STATE

We provide the first PC traversal for cryoDRGN in Figure 51. CryoDRGN does not recover the upper part of the protein at all. In addition, there is no motion through the principal component. This might indicate that the recovered motion is in fact noise in the top part. We additionally plot 3 structures taken evenly along the first principal component of the cryoStar volume method in Figure 52. The method is also unable to reconstruct the very mobile top part, in spite of using the latent space of the structural method. For Pfr, the debiasing technique of cryoStar through a volume method is ineffective for the top part.

Finally, we run cryoStar volume method on the latent space recovered by cryoSPHERE and show the first principal component traversal in Figure 53. Similar to cryoDRGN and cryoStar Phase II, this procedure is unable to reconstruct the top part of the protein. Hence this allows debiasing on the bottom part of the protein only.

We provide a movie of the first PC traversal of cryoStar volume method with cryoSPHERE latent variables here and the same movie with cryoStar latent variables here.

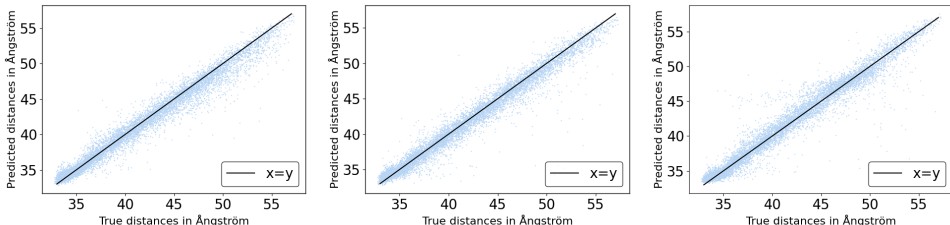

Figure 42: MD dataset, SNR 0.1. True versus predicted distances. From left to right: 10 segments, 20 segments, 25 segments.

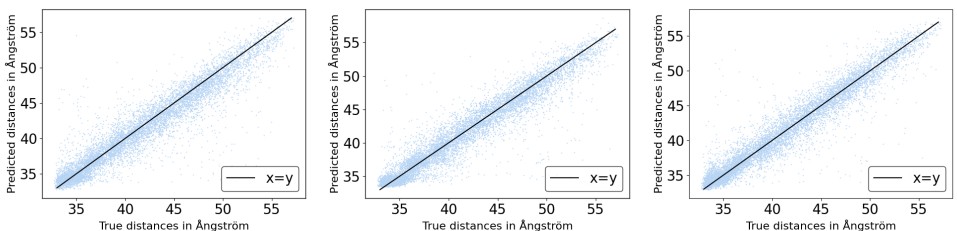

Figure 43: MD dataset, SNR 0.01. True versus predicted distances. From left to right: 10 segments, 20 segments, 25 segments.

### B.4.2 PR STATE

We provide three structures taken evenly along the first principal component of cryoDRGN in Figure 54. There is no motion in the bottom part as expected. However, the method is unable to recover the top part.

We also run cryoStar volume method on the latent space recovered by cryoSPHERE and display the first principal component traversal in Figure 56. We encounter the same difficulties as cryoStar volume method and cryoDRGN, see Figures 54, 55. CryoDRGN is not able to recover the top part of the protein. While cryoStar volume method does recover some motion for this top part, the resolution is too low to debias the base structure. Therefore, this debiasing procedure is only useful for the lower part of the protein.

We provide a movie of the first PC traversal of cryoStar volume method with cryoSPHERE latent variables here and the same movie with cryoStar latent variables here.

### B.5 COMPUTATIONAL COSTS

CryoStar and cryoSPHERE share the same way of turning a structure into a volume. This is a computationally expensive procedure, reduced by the sperability of Gaussian kernels, see e.g (Chen et al., 2023).For an of size $N_{\text{pix}} \times N_{\text{pix}}$, it involves computing two times the distance of each residues to $N_{\text{pix}}$ pixels and taking the product of these vectors to obtain a matrix representing the images for each residue, then summing over the residues. This is one of the computational bottlenecks of the structural methods. In spite of sharing the same bottleneck, cryoSPHERE tends to be slightly more computationally demanding. This is because cryoSPHERE needs to compose $N_{\text{segm}}$ rotations for each residue, while cryoStar only translates each residue. For example, for the experiments of Section 5.3, in 24 hours, cryoStar performs roughly 190 epochs and cryoSPHERE with $N_{\text{segm}} = 20$ performs roughly 130 epochs. Similarly, for the experiment of Section 5.2, cryoStar performs 184 epochs while cryoSPHERE with $N_{\text{segm}}$ performs 95 epochs. However, owing to the reduced number of freedom of cryoSPHERE compared to cryoStar, we observe that cryoSPHERE performs as well or better as cryoStar for the same computational budget.

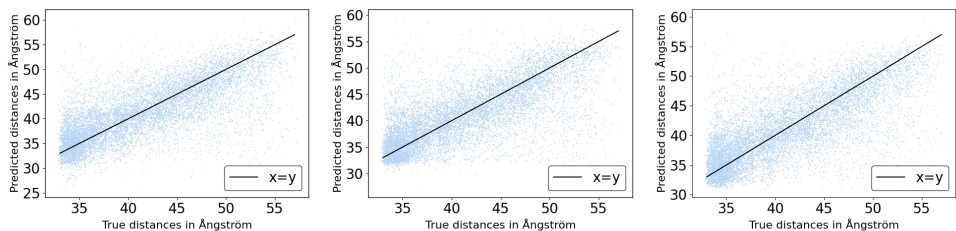

Figure 44: MD dataset, SNR 0.001. True versus predicted distances. From left to right: 10 segments, 20 segments, 25 segments.

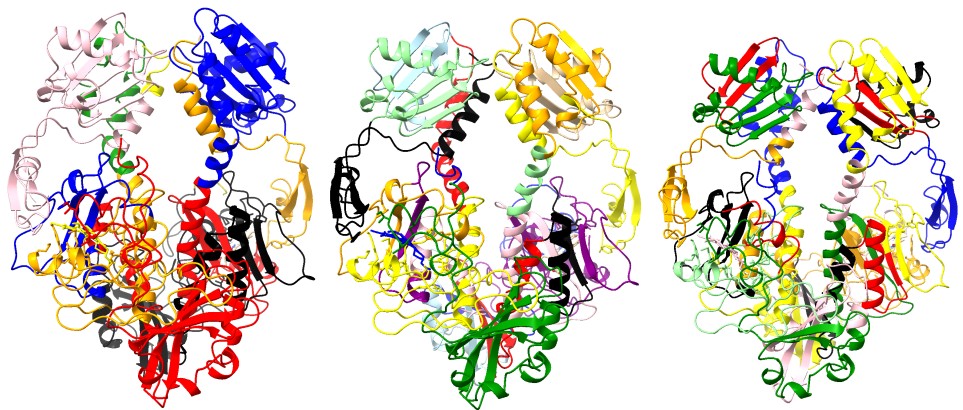

Figure 45: MD dataset, SNR 0.001. Segments decomposition. From left to right: 10 segments, 20 segments, 25 segments.

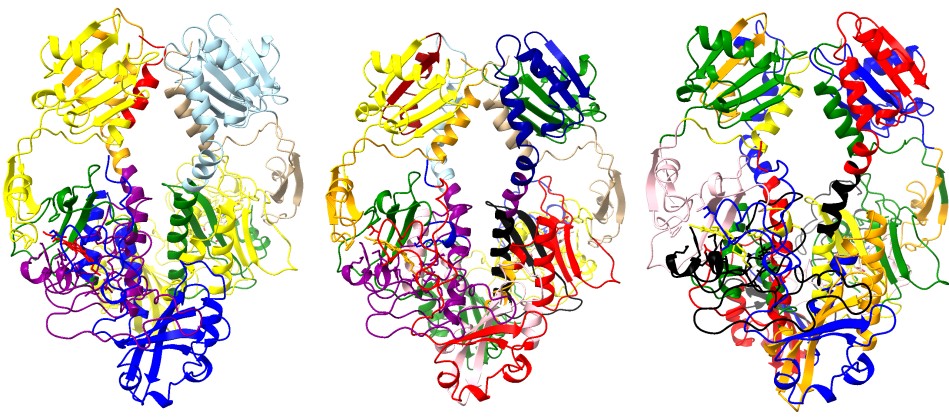

Figure 46: MD dataset, SNR 0.01. Segments decomposition. From left to right: 10 segments, 20 segments, 25 segments.

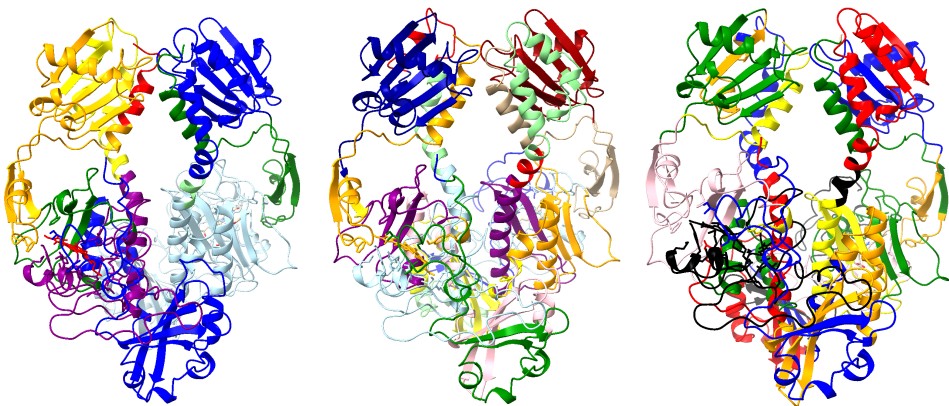

Figure 47: MD dataset, SNR 0.1. Segments decomposition. From left to right: 10 segments, 20 segments, 25 segments.

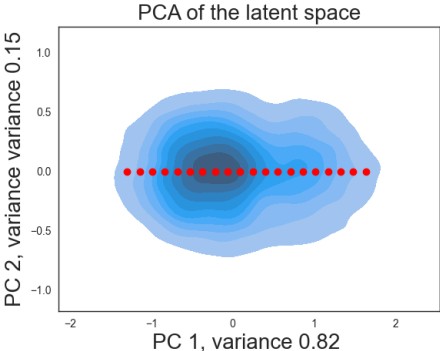

Figure 48: Empiar 10180: kernel density plot of the first and second principal components of that latent space of cryoSPHERE. The red dots are the point selected for the traversal of the first principal component.

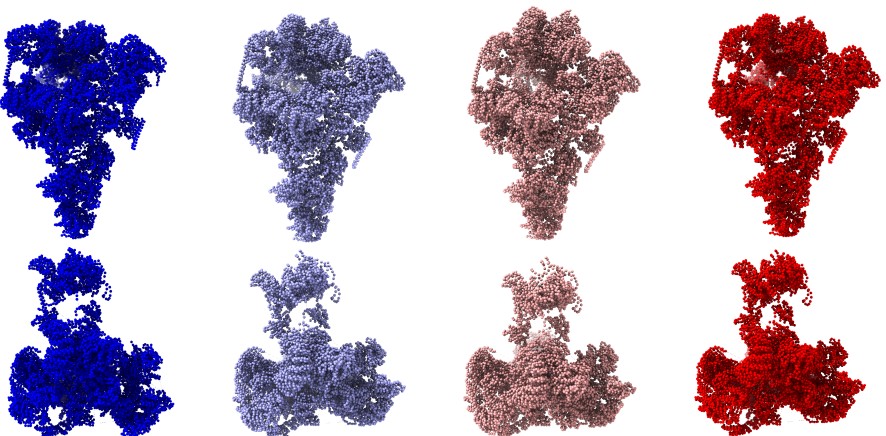

Figure 49: Empiar 10180. Four structures taken along the first principal component, from blue to white to red. Top: view from the "back". Bottom: view from the "top".

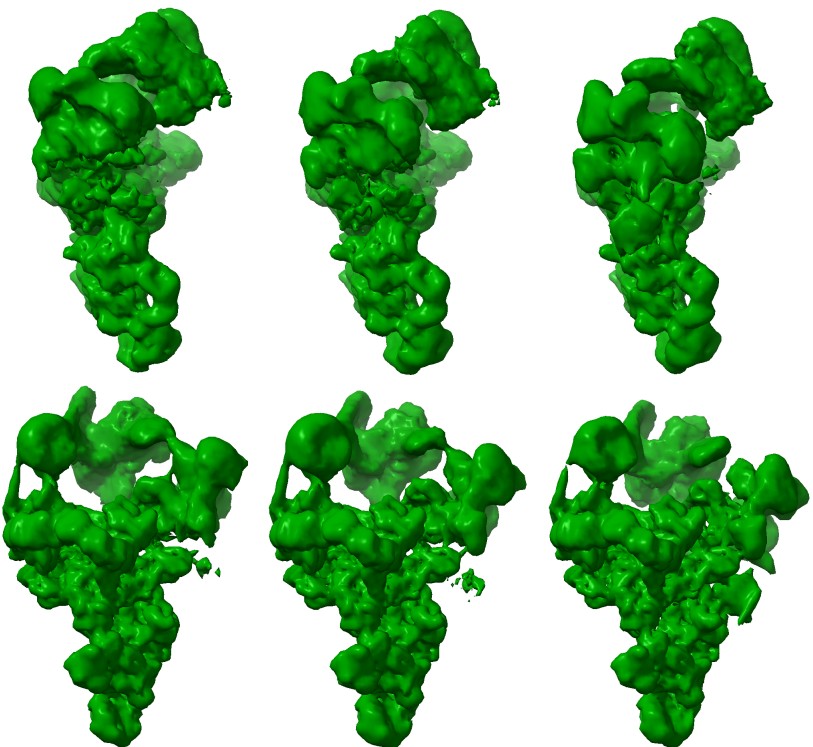

Figure 50: Empiar 10180, DRGN-AI is trained on the latent space of cryoSPHERE. Three Volumes taken evenly along the principal component.

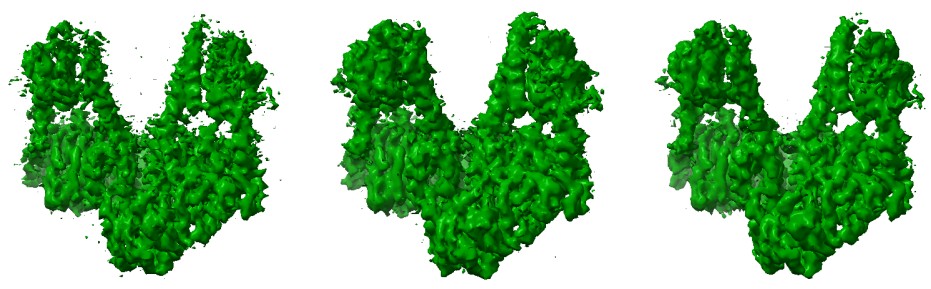

Figure 51: Pfr: 3 volumes taken evenly along the first principal component of cryoDRGN.

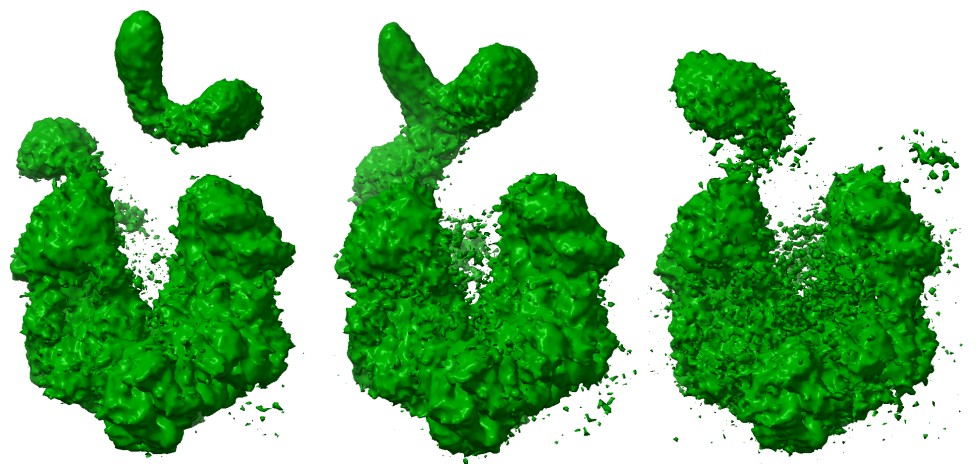

Figure 52: Pfr: 3 volumes taken evenly along the first principal component of cryoStar volume method.

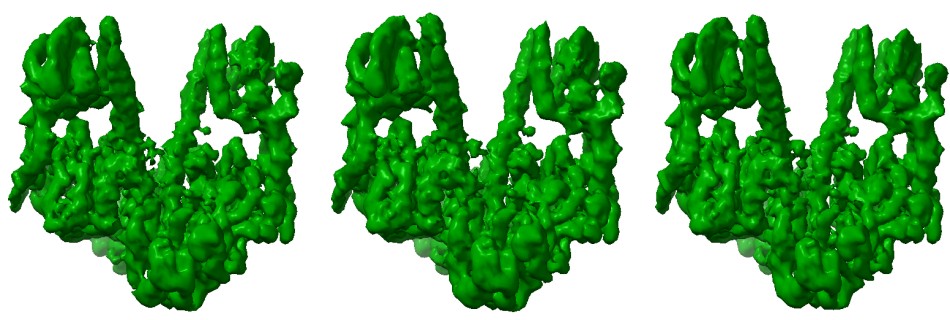

Figure 53: Pfr state. CryoStar volume method is trained on the latent space of cryoSPHERE. From left to right: three volumes taken evenly along the first principal component.

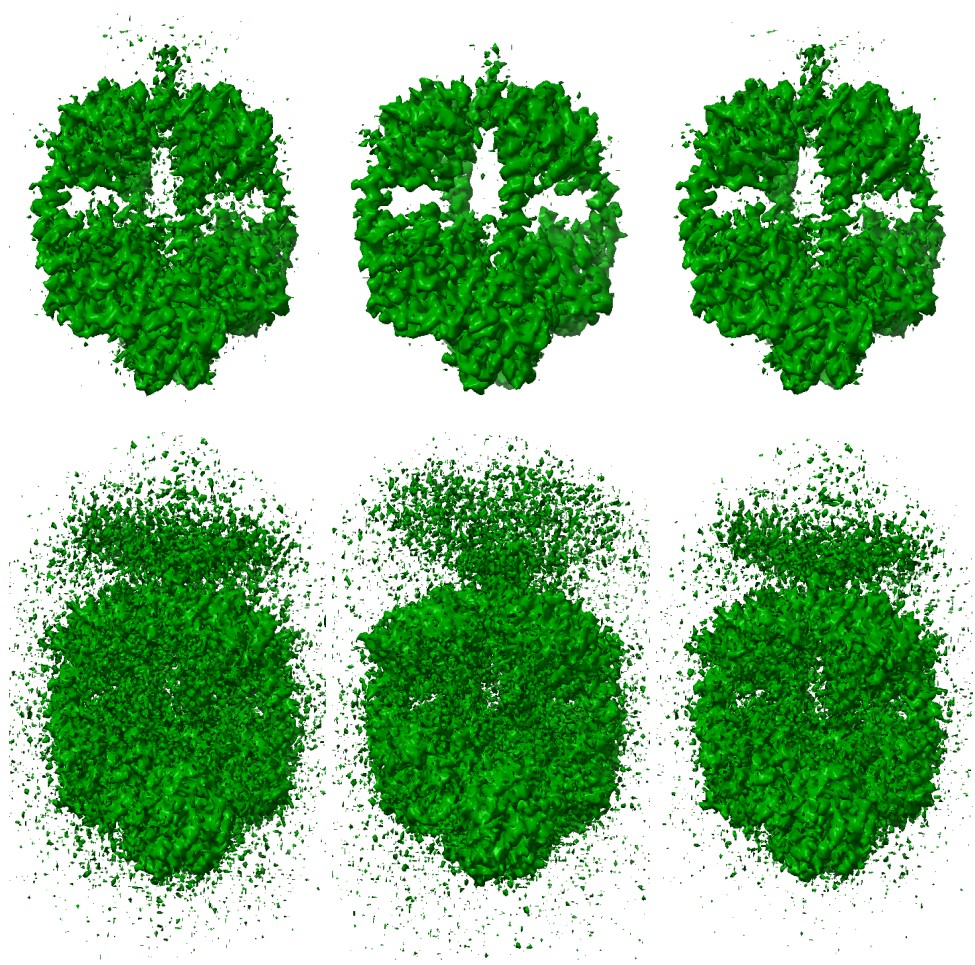

Figure 54: Pr: 3 volumes taken evenly along the first principal component of cryoDRGN volume method. The top and bottom volumes are the same with a different density threshold.

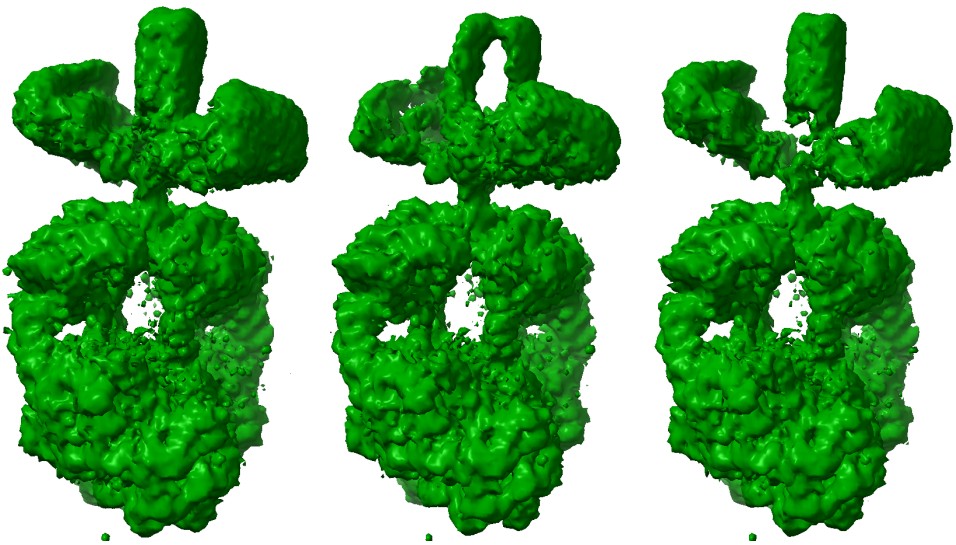

Figure 55: Pr: 3 volumes taken evenly along the first principal component of cryoStar volume method.

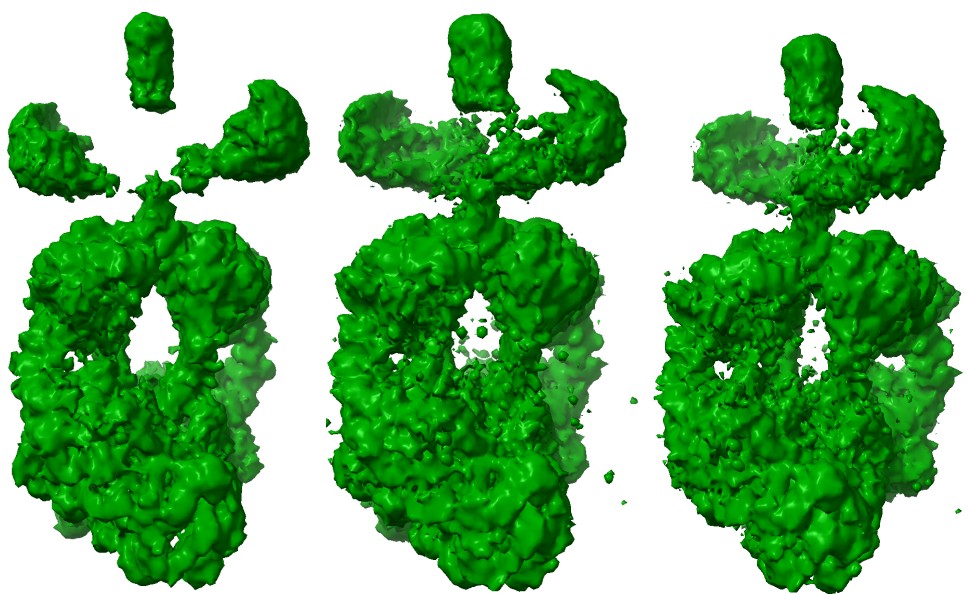

Figure 56: Pr state. CryoStar volume method is trained using the latent space of cryoSPHERE. From left to right: volumes taken evenly along the first principal component.