# OpenReview forum: "cryoSPHERE: Single-Particle HEterogeneous REconstruction from cryo EM"
_ICLR.cc/2025/Conference — ICLR 2025 Poster_

### Official Review · Reviewer_JGA5 · 2024-10-30

**Soundness:** 3
**Presentation:** 2
**Contribution:** 3
**Rating:** 6
**Confidence:** 4

**Summary:**

This paper introduces cryoSPHERE, a deep learning model that uses nominal protein structures (e.g., from AlphaFold) to break proteins into segments and fit these segments to various conformations in cryo-EM data.

**Strengths:**

1. The approach of learning to decompose the protein’s amino acid chain into segments to represent various conformations is novel.
2. The interpretability of moving part is highly beneficial for practitioners, offering a clearer understanding of conformational dynamics.

**Weaknesses:**

1. No meaningful improvements were observed over CryoStar
    - Figures 3, 12, 13: It’s unclear what meaningful differences exist compared to CryoStar.
    - Figures 14, 21: There doesn’t appear to be a significant difference between CryoStar and CryoSphere (CryoStar may even appear better).
    - Figure 29: It would be great to compare CryoStar with CryoSphere rather than with CryoDRGN.
2. Should change Figure 6:
    - Instead of showing only CryoDRGN results and G.T, a comparison with CryoSTAR or Dynamight would be more appropriate (perhaps include Figure 32 and the CryoSphere result, as there are no CryoSphere results with SNR 0.001).

**Questions:**

Could you include additional comparisons with methods like CryoSTAR, or Dynamight (e.g., computational cost, novelty, etc)

**Details Of Ethics Concerns:**

No concerns

---

> ### Author Response · Authors · 2024-11-20
> **Weakness 1**
>
> We are thankful to the reviewer for his thorough reading of our figures and for requesting missing information/figures. We are going to add them.
>
> 1/While the recovered distances look the same for both cryoStar and cryoSPHERE the FSC are in favor of cryoStar (Figure 4, 15 and 22), especially for the lower SNR cases. This indicates that  cryoSPHERE is less prone to overfitting thanks to its local rigidity through segments.
> Importantly, we now provide a new analysis on an experimental data set. The protein has two states, where one is significantly more disordered. Analysis of the heterogeneity in this data set shows that cryoStar notably overfits the disordered state, leading to clashes in the protein structures, while cryoSPHERE does not do this. See also the general answer.
>
> 2/In the high SNR data set (Fig. 15), cryoStar is very marginally better (<< standard deviation at the 0.5 cutoff), in the intermediate SNR (Fig. 21) the performances of cryoSPHERE are marginally better (Fig 22) and for the low SNR dataset (Fig. 4), cryoSPHERE outperforms cryoStar by one standard deviation. These figures fully support our claim that cryoSPHERE is more resilient to noise than cryoStar.
>
> The general limitation underlying all efforts to fit structures directly to cryo EM images is the limited content of information given low S/N levels. cryoSPHERE offers a major advance over cryoSTAR since the amount of structural parameters to be refined against the images are vastly reduced.
>
> This is a dramatic difference when it comes to experimental data. In the revised version we plan to include analysis of an experimental cryo EM data set. See the general response. CryoSTAR leads to overfitting/physically unrealistic deformation of the protein for the case where a large degree of conformational heterogeneity is present, while CryoSPHERE is robust to this.

---

> ### Author Response · Authors · 2024-11-20
> **Weakness 2**
>
> We included Figure 6 in the main paper to motivate the need for structural methods in case of very noisy datasets. However, it is true that since cryoStar is the most similar method, it would have made more sense to compare to cryoStar. The poper has undergone substantial change now.

---

> ### Author Response · Authors · 2024-11-20
> **Questions**
>
> As explained in the general answer, we have added a new experimental dataset, untackled so far, which demonstrates that cryoSPHERE recovers physically plausible results even for experimental data sets with high conformational heterogeneity. This is important, because it pushes the applicability of the methods towards more complex biological problems. See section 5.3.
> Regarding e2gmm and dynaMight we are doing our best to provide a comparison, see general answer.
> Finally, we have added a discussion of the computational costs of cryoStar and cryoSPHERE in appendix B.5

---

### Official Review · Reviewer_NciV · 2024-10-31

**Soundness:** 3
**Presentation:** 3
**Contribution:** 3
**Rating:** 8
**Confidence:** 4

**Summary:**

- The authors propose a VAE and GMM-based approach to atomistic heterogeneous Cryo-EM reconstruction
- The GMM part learns how to divide the amino acid chan in segments
- The VAE learns how to encode images into latent variables and decode these variables along with the GMM output into segment-wise conformational changes
- The authors claim that their method overcomes current challenges that atomistic models face, while also achieving improved interpretability by learning these segments (that ideally correspond to domains in the biological sense)

**Strengths:**

The work is of high quality and overall very well-written. I’d like to highlight the following strong points of the paper:
- The authors seem very well aware of the difficulties atomistic models have in terms of the optimisation landscape and address the problem adequately by regularising while keeping sequential information intact
- The authors give useful biological remarks such as that their methods seems to recover domains, which is useful to know as a practitioner
- The authors show that their method works on real data, which is actually the first time I’ve read this for atomistic models

**Weaknesses:**

There is one main weak point in the paper in my regard:
- The authors do not address limitations of their method very thoroughly:
    - whether or not their method is able to overcome poor initialisation of the initial structure S_0. This is one of the major risks of working with (pseudo)-atomistic models as shown in works such as DynaMight (Schwab et al. 2024)
    - Whether or not local physical information such as bond lengths between atoms at the boundaries of the segments is preserved. This might be challenging

**Questions:**

Questions for clarification
- In section 3.6 the authors say “Similar to Li et al. (2023), instead of using a mean squared error loss between the predicted and ground truth image, we use a correlation loss between the true and predicted image”. However, they do not motivate this choice. Would you please care to elaborate?
- In section 5.1 the authors write “Testing the segment decomposition, we then run cryoSPHERE by requesting division into Nsegm = 4. The program learnt a first and third segment with 0 residues, a second segment with 1353 residues and a fourth segment with 157 residues (Figure 5). Thus, cryoSPHERE learnt segments according to the ground truth.” Is there a reason why we should expect that the model prefers to learn two segments?

Additional feedback
- It reads as if equation 7 is the only loss that is being used. I assume that the VAE latent distribution is also trainable? It would benefit the readability if the authors clarify whether the loss is just the decoder loss or the whole loss.“
- In section 5.3 the authors write “The first two principal components explain more than 96 percent of the variance.” In general, I would be very reluctant to make any claims about PCA in the latent space as you could come up with a transformation that scrambles the latent space and still gives the same decoding, but covers much less variance. I can imagine that Figure 7 is a nice illustration, but I would recommend to leave it with that.

---

> ### Author Response · Authors · 2024-11-20
> **Weakness 1**
>
> We want to thank the reviewer for acknowledging the strengths of our approach.
>
> We are aware that poor initialization of the S_0  can be hard to overcome. This is why it is important to verify the model bias with a volume method. As explained in our answer to reviewer RJxo, weakness 3, a run of DRGN-AI after cryoSPHERE is similar to the cryoStar volume method at almost no extra cost. This can help alleviate the bias problem. In our experience, cryoSPHERE still works when parts of the protein are missing in the base structure or the structures present in the dataset.
> In a next version of our rebuttal draft, we plan to show an experiment where our atomic model has too many residues and a bias check with DRGN-AI.
>
> Regarding the segmentation: in our experience, and upon visual inspection of the results (in the paper see e.g Figures 16, 24, 30) the segmentation does not break the protein at the boundaries of the segments. This holds without even using a structural loss for many datasets in our experience. However, for the more challenging cases like the experimental dataset we added in our rebuttal draft (see answer to reviewer RJxo, weakness 3) the structural loss defined in equation (9) and (10) is necessary. In that case the protein does not break apart.
> The smoothness of the transformation comes from the smoothness of the segmentation: we do not use any “hard” decomposition of the protein, therefore there are no real boundaries between domains.

---

> ### Author Response · Authors · 2024-11-20
> **Questions**
>
> 1/ Very often, real datasets are preprocessed resulting in arbitrary scaling (using e.g CryoSparc, Relion etc…) before running heterogeneous reconstruction methods. For example, the images can be normalized/standardized differently depending on the tools used. Using the mean square, this would prevent any successful reconstruction with a method such a cryoStar or cryoSPHERE, however, when using the correlation this problem does not occur. Note that volume methods such a cryoDRGN do not have this problem, since the magnitude of the predicted images does not rely on the values of a projected GMM. We have added a short explanation of this in section 3.6.
>
> 2/To our knowledge, there is no reason why the model would prefer two segments. This purely comes from the fact that the optimal deformation is the ground truth in such a simple case. But in more realistic settings, the motion does not happen in terms of rigid body motion of distinct segments. We do not expect the model to learn the optimal segmentation plus transformations that minimizes the number of recovered segments.
>
> 3/ Yes, the latent distribution is trainable. Thank you for pointing out this confusing equation. We have made this more explicit with equation (8)
>
> 4/We agree with this reasoning; using PCA to picking up structures from the latent variable is indeed only for displaying purposes. We have removed the text from the paper.

---

### Official Review · Reviewer_RJxo · 2024-11-04

**Soundness:** 2
**Presentation:** 2
**Contribution:** 2
**Rating:** 6
**Confidence:** 5

**Summary:**

The authors introduced a new method, cryoSPHERE, that resolves the continuous heterogeneity problem in cryo-EM reconstruction. Similar to some other methods, cryoSPHERE uses an atomic model as the input, makes segmentations, and uses the results as the regularization in finding the flexible movement in the cryo-EM dataset.

**Strengths:**

Instead of finding the deformation for each residue, cryoSPHERE learns a segmentation of the amino acid or nucleotide chains and deforms the segment to fit the heterogenous cryo-EM data. This is indeed a valid assumption in many cases especially for large complexes  like the spliceosome (EMPIAR-10180).

**Weaknesses:**

Major:
1. CryoSPHERE uses an atomic model as the reference and formulate the conformational heterogeneity in cryo-EM dataset as the deformation of the segments of the reference model. In finding the optimal segments, the approach is similar to what used in e2gmm, where a GMM with N_{segm} components is fitted. This segmentation result is then used in a cryoSTAR/DynaMight-like setting, replacing the regularization losses and compute predicted projections to compare with the particle images. The author should also compare the result with other similar methods, including DynaMight, e2gmm, and maybe 3DFlex, and discuss the outcomes.

2. In cryo-EM, density maps are extremely important in evaluating the validity of the results. However, the authors only presented the density maps for one of the synthetic dataset. Especially in the case of under the hypothesis that the heterogeneity is modeled as the deformations of a canonical density, it is very important that the reconstructed canonical density is free from any obvious artifact.

3. The FSC comparison in Sec. 5.2 between cryoSPHERE and cryoDRGN and cryoSTAR may not be very fair, since cryoSPHERE's density is reconstructed under the assumption that there exist a canonical density, while cryoDRGN does not hold the same hypothesis, and cryoSTAR explicitly avoids this to reduce the unwanted bias from the reference model. A fair comparison would be made against DynaMight and 3DFlex.

4. Lack of ablation study. The authors should discuss how N_{segm}, which is a key hyperparameter that is related to the degree of freedom in the segmentation, affects the results in both synthetic and real data.

5. The authors compare cryoSPHERE to cryoDRGN and cryoSTAR, but from the experiments, it is unclear what the advantage that cryoSPHERE brings to the field. The only advantage seems to be the increase FSC resolution, but it is only shown in one synthetic dataset.

Minor:
1. The sequence of Fig.6 and Fig.7 in the paper is reversed.
2. In the related works, e2gmm should also be considered as a method using deep learning.
3. In the FSC comparison, if not comparing two half maps, the cutoff of 0.5 should be used and 0.143 is not meaningful. See [1] appendix for the reason (and also why 0.143 is the number widely used for half maps comprison). Therefore the discussion at the end of Sec. 5.2 about 0.5 and 0.143 cutoff improvement is not very correct.
4. The writing could be better polished.
5. I do not think the "Ethics Statement" content is proper as it lacks evidence.


[1] Rosenthal, Peter B., and Richard Henderson. "Optimal determination of particle orientation, absolute hand, and contrast loss in single-particle electron cryomicroscopy." Journal of molecular biology 333.4 (2003): 721-745.

**Questions:**

Can the authors elaborate how the density maps (volumes) are reconstructed in cryoSPHERE (like in Fig. 6)? I understand how backprojection in cryo-EM works, but still find it difficult to understand how this is performed exactly.

---

> ### Author Response · Authors · 2024-11-20
> **Major weakness 1**
>
> We want to thank the reviewer for his thorough reading and the thoughtful parallels he/she makes with the rest of the literature.
>  We now reply to his comments.
>
> Point 1:
> We do understand the parallel between the N_{segm} of cryoSPHERE and the M of e2gmm. However, e2gmm and cryoSPHERE are very different on that respect:
>
> 1/ The rows of the NxM matrix relating the big GMM to the small GMM are not summing to one. Therefore, they do not represent the probability of belonging to the modes of the small GMM.
>
> 2/ The NxM matrix does not define an approximately rigid motion of parts of the protein. It is, instead, much more similar to the 3DFlex deformation field. To cite the e2gmm paper[1]:
>
> “As such, by multiplying a set of vectors that describe the movement of the small GMM with this matrix, each of the vectors can drive the movement of Gaussian functions in a local region from the large GMM, enabling the coordinated movement of the two GMMs (Figure 2(A-D)). Conceptually, the small GMM serves a similar role to the “deformation flow field” in the 3DFlex implementation”
>
> 3/The rigid body motion of e2gmm happens in a second round of training and needs user input segments. On the contrary, cryoSPHERE learns the rigid body motion and the segmentation in an end-to-end fashion, with only N_{segm} as input.
>
> This point being clarified, it is true that e2gmm accepts a base structure. Following the reviewers suggestion, we trying to install eman2 to compare e2gmm with cryoSPHERE. We will update our rebuttal when the results are available.
>
> [1] Integrating Molecular Models Into CryoEM Heterogeneity Analysis Using Scalable High-resolution Deep Gaussian Mixture Models, Chen et al.

---

> ### Author Response · Authors · 2024-11-20
> **Major weakness 2**
>
> We show the FSC to the ground truth for three different datasets, each of which with different SNR, see main body and appendices. In addition, the canonical structure we use for running cryoSPHERE is different from the one used to create the dataset.
> However, it is true that the structure approach alone cannot detect model bias. Practitioners usually run more than one method such as cryoDRGN or 3DFlex. For example, running DRGN-AI after cryoSPHERE comes at no extra cost compared to the volume method of cryoStar and will help detect potential bias in cryoSPHERE results. Running DRGN-AI on top of cryoSPHERE is similar to running cryoStar volume method on top of cryoStar structure method (see answer below), therefore, we did not implement any volume method in cryoSPHERE.
> In a future version of our rebuttal paper, we will provide a new dataset analyzed with an atomic model containing too many residues, and use DRGN-AI to detect the bias.

---

> ### Author Response · Authors · 2024-11-20
> **Major weakness 3**
>
> We do agree that cryoDRGN does not benefit from the information brought by a base structure. We provided a comparison to cryoDRGN to show that adding information in the form of an atomic model can provide better results, especially when the level of noise is high. In addition, for the comparisons in Sec 5.2, we would like to point out that:
>
> 1/Since cryoSPHERE and cryoSTAR use a base structure different from the ones on the images, the poses can only be considered approximate. This is not the case for cryoDRGN. In this respect the comparison is unfair to cryoSPHERE and cryoSTAR.
>
> 2/ Enforcing local rigidity of a base structure could negatively impact the 0.5 FSC cutoff compared to volume methods. The comparison with cryoDRGN is necessary to show that deforming a base structure can indeed improve the performances compared to volume methods.
>
> In addition, the cryoSTAR volume method (Phase II of cryoSTAR training)  is of little added value. We explain why in a 4 teps reasoning:
> 1/CryoSTAR is not end-to-end differentiable. Their volume method amounts to running DRGN-AI [2].
> See the section “Generating density maps with the volume decoder” in the method appendix of the paper CryoSTAR paper [3].
>
> 2/Hence, in a known pose setting, cryoSTAR and cryoDRGN share the same computational bottlenecks. One might as well run DRGN-AI instead of cryoStar volume method, unless fixing the latent space obtained with the structural method gives a resolution boost to the cryoStar volume method.
>
> 3/  Figures 15, 22, 29 of our paper demonstrate that the cryoStar volume method performs similarly or worse than cryoDRGN in terms of FSC. To our knowledge, we are the first ones to assess the volume method of cryoSTAR.
>
> From this we conclude that if one wants to detect model bias in structural methods, running the cryoStar volume method has no added value compared to running DRGN-AI before or after the cryoStar structural method.  Hence, we can also detect model bias in cryoSPHERE using cryoDRGN, at little extra cost compared to cryoStar structural + volume methods.
>
> [2] Revealing biomolecular structure and motion with neural ab initio cryo-EM reconstruction, Levy et al
> https://www.biorxiv.org/content/10.1101/2024.05.30.596729v1
>
> [3] CryoSTAR: leveraging structural priors and constraints for cryo-EM heterogeneous reconstruction, Li et al, https://www.nature.com/articles/s41592-024-02486-1

---

> ### Author Response · Authors · 2024-11-20
> **Major weakness 4**
>
> We agree and in our updated draft, we now provide runs of cryoSPHERE with N_{segm}=10, 20, 25 in appendix B.2.5 and show that the FSC increases with the number of segments. This will improve the quality of the paper, thank you for having pointed that out.

---

> ### Author Response · Authors · 2024-11-20
> **Major weakness 5**
>
> We show FSC curves for 3 different datasets, with various levels of noise and observe a better FSC relative to the other methods with decreasing SNR. In addition, the segmentation plus rigid body transformation approach provides more insights into the dynamics of the different parts of the protein compared to a per residue approach, something cryoStar does not provide. The new experimental dataset in our new draft shows that cryoStar structural method is unable to recover physically plausible results, and their volume method is not able to reconstruct the top part of the protein. CryoSPHERE, on the contrary, produces physically plausible results.
> In addition, we have now ran cryoSPHERE with N_{segm} = 25 and show that we perform very marginally worse than cryoStar on SNR 0.1, better on SNR 0.01 and one standard deviation better on SNR 0.001. See Figures 4, 15, 22.

---

> ### Author Response · Authors · 2024-11-20
> **Minor weaknesses**
>
> 1/ This part of the paper has completely changed now.
> 2/ Thank you for pointing out. We will change that.
> 3/ We agree with the review that the 0.5 threshold should be reported when comparing volumes to the ground truth. However, this is true under assumption that the recovered volume writes as true_volume + noise and we believe that this assumption is broken for structural methods. Hence, we chose to report both cutoffs to represent the behaviour of structural methods at different scales. But ultimately, one should look at the entire FSC curve, and that is why we reported them.
> 5/ We removed it.

---

> ### Author Response · Authors · 2024-11-20
> **Questions**
>
> In brief, the backprojection algorithm is the application of the Fourier slice theorem: given a set of images and their associated poses, we can reconstruct slices of the Fourier transform of the volume. This is, of course, a homogeneous method. CryoDRGN can be seen as a generalisation of this back projection algorithm for tackling heterogeneous data.

---

> ### Author Response · Authors · 2024-11-25
> **Discussion period ending soon.**
>
> Dear Reviewer,
> The discussion period ends in two days and we feel that we have now taken into account your suggestions. See our updated draft,  the general answer and the specific answers we sent you.
> We would love to hear your opinion on these updates.
>
> Best,
> The cryoSPHERE authors.

---

> ### Comment · Reviewer_RJxo · 2024-11-25
>
> I would like to thank the authors for the detailed explanation and the effort to make the manuscript more solid. Many of my concerns were adequately addressed. However, in real life, people care about and trust the densities with motion much much more than the "structure", a.k.a the atomic models, due to validation reasons. The authors should include at least the density maps with motion using cryoSPHERE on the real datasets (both EMPIAR-10180 and EMPIAR-12093). I fully agree with that cryoSTAR's volume decoder is cryoDRGN "shared the same bottlennecks" and DRGN-AI should improve the resolution of the density maps in this setting. The authors may choose either the using the cryoDRGN/cryoSTAR volume approach or DRGN-AI for the density generation. In other words, I believe that the key results in this paper should be density maps with motions being solved for real datasets by cryoSPHERE, and I believe those are still missing (apologies in advance if I failed to find them instead).
>
> I will keep my score for now, but would be happily raise the score if the densities with motion, fitting by cryoDRGN/cryoSTAR/DRGN-AI after cryoSPHERE, are shown and the performance is reasonably well.

---

> > ### Author Response · Authors · 2024-11-27
> > **Our improvements based on your suggestions**
> >
> > Dear reviewer,
> >
> > We thank you for this final rebuttal version of our paper. We have now computed and implemented debiasing volumes for both of the experimental data sets (see figure 5 and 6). In 2 out of 3 cases, we found excellent agreement.
> >
> > In detail, we find that for the splicosome (Figure 5) the reconstructions by cryosphere (and cryoSTAR) were mostly unbiased, except for a domain called U2, which appears to become totally disordered as identified by the new analysis.
> >
> > For the Pr state of the phytochrome (figure 6), the reconstructions were mostly unbiased (confirmed by the new analysis).
> > For the Pfr state of the phytochrome (figure 6), the unbiasing methods fail to produce meaningful densities for the upper part of the protein, due to the low signal levels in the unbiasing densities. This is despite the fact that cryosphere (but not cryoSTAR) can reconstruct meaningful structural ensembles for this state. We interpret this as a limitation in the debiasing method itself – this is an important area of improvement in future research into the subject.
> >
> > We discuss these points in the discussion and experimental section. We have also added more examples of volumes and movies of the motions recovered by the volumes methods on EMPIAR 12093, see Appendix B.4.
> >
> > We hope that the reviewer will agree to that adding these densities has made the reconstructions much more believable and even identified a limitation in the debiasing method itself, opening up for further research.
> >
> > Best,
> >
> > The authors of cryoSPHERE.

---

> > ### Author Response · Authors · 2024-12-02
> > **Discussion period ending soon**
> >
> > Dear reviewer,
> >
> > The discussion period ends soon and we believe that our last draft takes your suggestions into consideration. We now demonstrate how to obtain volumes based on cryoSPHERE in two different ways: using DRGN-AI and cryoStar volume method, as you requested. We apply this procedure to two real datasets, see sections 5.2 and 5.3. Let us know if you have further questions.
> >
> > Best,
> >
> > The authors of cryoSPHERE

---

> > > ### Comment · Reviewer_RJxo · 2024-12-02
> > >
> > > Thank you for adding the reconstructed density maps to the main figures and also in the appendix. While I have different opinions that the limitation is on the "debiasing method" itself, the addition of the density maps definitely make the paper more convincing on the application side. I also have concerns about the performance showing by the density maps on the EMPIAR-12093 dataset, as the motions resolved by the atomic models seem different than the density maps, which is difficult to interpret from an end user's stand point. However, the novelty introduced in this paper is reasonable and as a concurrent work of cryoSTAR, the performance is acceptable. I will raise my score as promised.

---

### Author Response · Authors · 2024-11-20
**General answer**

Here, we provide brief & general answers. We have submitted a new version with major revision and a new experimental data set, which directly shows that the approach used in cryoSPHERE outperforms other methods, in particular cryoSTAR, for experimental data of samples with high conformational heterogeneity.

Given the importance of devising robust and usable methods to analyze dynamics from cryo EM data, and the unique advantages of CryoSPHERE to cope with the high noise levels in real experimental data, we hope that the reviewers can agree that this is an important contribution to the conference.

None of the links we provide in this rebuttal refer to any accounts or identity of the authors of cryoSPHERE.

1/ We agree that comparison to other methods is important. In the revised version we planned to extend comparisons to e2gmm and dynaMight. However, dynaMight only produces bugs when run with a reference structure a often even without reference structure, thus it limits comparability, and we still cannot install eman2. We keep trying.

2/ Regarding comparison with CryoSTAR: cryoSTAR is concurrent work that has been developed independently from cryoSPHERE. In the new version of this paper we show more conclusively that CryoSPHERE is more resilient to noise compared to cryoSTAR (see fig. 4, see reviewer JGA5, point 1 for a detailed reply). This is expected because the amount of free structural parameters is much reduced in cryoSPHERE, which is highly relevant when interpreting experimental data sets.

3/ To demonstrate the applicability of cryoSPHERE to high levels of noise, we included in the revised version  the analysis of an experimental cryo EM data set (medium size protein recorded by a state-of-the-art experimental facility) in the revised version of the paper. Here, cryoSTAR leads to overfitting/physically unrealistic deformations with clashes of the protein for the “disordered state”, while cryoSPHERE gives plausible results. This is highly relevant, because the most interesting data sets typically have low SNR and high degrees of structural heterogeneity.

4/ Regarding the debiasing of the results: All methods that directly fit structures to the images face this challenge. cryoSTAR has implemented this as a phase II training (the volume method), however, we prefer to run cryoDRGN/DRGN-AI in a second, separate step of training. See our answer to reviewer NciV weakness 3. Figures 15, 22, 29 show this approach gives similar or better results to the implementation that cryoStar has chosen. In any case, when dealing with highly heterogeneous data sets (like the experimental data that we analyze), post computation of the volumes is only of limited value as the consensus densities are very low amplitude. We do agree that for all methods, more rigid methods for debiasing of the results should be developed.




Explanation of point 1/

We genuinely tried to compare cryoSPHERE to DynaMight. However, this comparison only makes sense if we can run it using a base structure, as in cryoSPHERE. This is one of the two modes in which DynaMight can be run. Unfortunately, the DynaMight documentation does not mention how to run it with a base structure, as you can see on the github page

 https://github.com/3dem/DynaMight
or on the official documentation

 https://relion.readthedocs.io/en/release-5.0/SPA_tutorial/Flexibility.html.

From the source code, we figured out how to use this option, see line 59:
https://github.com/3dem/DynaMight/blob/main/dynamight/deformations/optimize_deformations.py

However, when running this we only encountered bugs that led to the termination of the DynaMight training. This problem has been noted by others as you can see with this unanswered github issue (not raised by us!):
https://github.com/3dem/DynaMight/issues/11

In addition, neither the official documentation nor the github page explain how to obtain one structure/volume per image. They both describe how to display volumes based on a GUI. This is another obstacle to quantitatively compare DynaMight with cryoSPHERE.
We feel that we have genuinely tried to run DynaMight as a structural method and that it is very unfortunate that it was not working properly.
We thought it would be considered unfair to compare its results with cryoSPHERE without starting with the same base structure. But since the reviewers insist on DynaMight, we are currently trying to run it without base structure and will try to modify its source code to obtain one structure/volume per image, even though we also encounter bugs in this mode on many datasets. This will be used to make a DynaMight/cryoSPHERE comparison if we succeed in running it.
Regarding e2gmm: we have been consistently unable to install the latest version of eman2, in spite of following the documentation. We are still trying to the best of our efforts.

---

### Author Response · Authors · 2024-11-20
**What is new in the revised version of our draft:**

We want to thank the reviewers for their thoughtful comments and we feel that their suggestions have improved our paper. Here is a summary of what has changed:

1/ We have added a new experimental dataset, featuring a high level of noise and heterogeneity. We show that cryoSPHERE is able to recover physically plausible motion, while cryoStar overfits.

2/We change the run of cryoSPHERE with N_segm = 20 to N_segm = 25. This way we perform very marginally worse than cryoStar on SNR 0.1, marginally better than cryoStar on SNR 0.01 and one standard deviation better than cryoStar on SNR 0.001

3/ We provide a discussion of the impact of the choice of N_segm in appendix B.2.5.

4/ We also have change more minor points raised by the reviewers, see specific answers to the reviewers.

---

### Author Response · Authors · 2024-11-25
**Addition to the rebuttal**

We would like to thank the reviewers again for their suggestions. They have improved our paper a lot.

We upload a new revised version of our paper, this time adding Appendix B.2.4 showing that the Phase II of training cryoStar, to assess the bias, can also be performed with cryoSPHERE using DRGN-AI, at no extra cost.

We have been trying the best we could to run DynaMight and e2gmm. However:

1/ As explained in the previous general answer, we have not been able to run DynaMight with a base structure. (There seems to be some bug in the available implementation that prevents this. This has been observed by others as well, as indicated by the previously mentioned github issue.)

2/ We therefore ran the method without base structure on the synthetic datasets. Unfortunately, as far as we can tell, the only way to obtain the volumes is through a GUI which makes it impossible to obtain the 1k volumes that would be needed to compare DynaMight and cryoSPHERE quantitatively. In addition, the volumes recovered are low resolution, since the high resolution reconstruction is done in a second step, on a single volume for DynaMight. This prevents any meaningful comparison in terms of FSC.

3/ We only encountered errors when trying to run DynaMight, with and without base structure, on EMPIAR 12093.

4/After much efforts, we could finally install eman2. Unfortunately, when running e2gmm on any of our datasets, we encountered Out Of Memory errors terminating the training. We have tried many unsuccessful solutions: to reduce the box size, to use only the backbone of the base structure instead of the full pdb, to reduce the batch dimension to 8, to increase the number of GPUs to get more RAM, to run it on only 50k images, but nothing has worked.


We have genuinely tried to compare cryoSPHERE to e2gmm and DynaMight and we regret that we could not, due to issues with the publicly available code for these methods. We hope that the reviewers can acknowledge that providing new and stable solutions to the problem of extracting conformational heterogeneity is highly interesting.

---

### Author Response · Authors · 2024-11-27
**New rebuttal**

We thanks the reviewers again for this new rebuttal version of our paper. We feel that we have a much stronger case now and that their suggestions helped us precise the strengths and shortcomings of cryoSPHERE.

We now summarize the changes made :

1/ We now demonstrate two ways to debias cryoSPHERE: using DRGN-AI for EMPIAR 10180 and using cryoStar volume method for EMPIAR 12093 (sections 5.2 and 5.3 respectively), using cryoSPHERE's latent space.

2/ We have added examples of volumes showing this debiasing procedure, see Figures 5 and 6.

3/ We have added more examples of volumes and movies of the motions recovered by the volumes methods on EMPIAR 12093, see Appendix B.4.

4/ We now conclude that we can debias cryoSPHERE successfully on real data, but that for the most noisy and heterogeneous datasets, volume method debiasing is still impossible (for cryoSPHERE and cryoStar). We leave that problem to future work.

---

### Meta-Review · Area_Chair_PWKK · 2024-12-16

**Metareview:**

The paper proposes cryoSPHERE, a method for reconstructing proteins that can occur in different configurations.

The paper received three reviews, and after the rebuttal all three reviewers are supportive of publication.

* Reviewer RJxo was initially not supportive of publication; the reviewer noted that that some comparisons are not fair, and pointed out that it is unclear what the advantage of the method is over methods like cryoSTAR and others. In the rebuttal, the authors highlight that cryoSPHERE can provide physically plausible reconstructions for examples where cryoSTAR cannnot. After the rebuttal, the reviewer still has concerns about the performance of the method, but finds the novelty acceptable to recommend weak accept.

* Reviewer NciV is supportive of publication and highlights that the method is evaluated on real data, contrarily to other works in the area that are based on atomistic models.

* Reviewer JGA5 finds elements of the approach to be novel and the interpretability of the moving part beneficial for understanding conformational dynamics of proteins. However, the reviewer also notet that there are no meaningful improvements over CryoSTAR. In the rebuttal, the authors provided new figures that partially addressed the reviewers concerns, and while the reviewer still has concerns about the novelty relative to CryoSTAR, the reviewer raised the score to weak accept.

The paper provides a new method for reconstructing proteins in different configuration, and demonstrates that the method has, in some cases, a slight benefit over existing methods, in particular over cryoSPHERE. I agree with the reviewers that while the method uses many elements from prior methods (such as the way the VAE is used, the atomistic representations, etc) and while the benefit over existing methods is relativly marginal, the method and evaluations contribute to the important problem of reconstructing a single protein in different configurations, and I therefore recommend to accept the paper.

**Additional Comments On Reviewer Discussion:**

see above

---

### Decision · Program_Chairs · 2025-01-22

Accept (Poster)